# Single cell proteomic analysis defines discrete neutrophil functional states in human glioblastoma

Pranvera Sadiku[1,10], Alejandro J. Brenes [1,10] ✉, Rupert L. Mayer [2,3,4], Leila Reyes[1], Patricia Coelho[1], Gabi van Stralen[1], Ailiang Zhang[1], Manuel A. Sanchez-Garcia [1], Emily R. Watts[1], Imran Liaquat[5,6], Andrew J. M. Howden [7], Ikeoluwa Adekoya[8], Anuka Boldbaatar[8], Allan MacRaild[8], Sarah Risbridger [8], Gillian M. Morrison[5], Heather MacPherson[5], Caroline M. Bruce [1], Shonna Johnston[1], Robert Grecian[1], Fiona A. Murphy[1], Steven M. Pollard [5,6], Paul M. Brennan [9], Karl Mechtler [2,3,4] ✉ & Sarah R. Walmsley [1] ✉

Neutrophils are vital innate immune cells shown to infiltrate glioblastomas, however we currently lack the molecular understanding of their functional states within the tumour niche. Given that neutrophils are known to display a prominent discordance between mRNA and protein abundance, we developed ultra-sensitive mini-bulk and single cell proteomic (SCP) workflows to study the heterogeneity of peripheral blood and tumour associated neutrophils (TAN) from patients with glioblastoma. Mini-bulk analysis enabled a deeper protein coverage of circulating immature, mature and TAN populations, defining signatures of maturity and demonstrating that TANs resemble mature circulating neutrophils. Analysis of the SCP data results in the detection of >1100 proteins from a single TAN providing a detailed characterization of neutrophil subsets in glioblastoma. Our approach shows evidence of pathogenic and anti-tumorigenic clusters and discovers cell states invisible to scRNAseq, opening new opportunities to selectively target pro-tumoural neutrophil states.

Glioblastoma (GBM) is a grade IV glioma, the most common and aggressive primary brain cancer in human adults[1]. The current standard of care for affected patients consists of treatment involving a combination of maximal surgical resection with radio- and chemotherapy. Despite treatment, prognosis is very poor with a 5-year patient survival rate of 4%[2]. With extensive neutrophil infiltration into GBM tumours reported, neutrophils have the potential to present themselves as potent therapeutic targets[3]. To effectively target neutrophils in GBM we

[1]Center for Inflammation Research, Institute for Regeneration and Repair, University of Edinburgh, Edinburgh, UK. [2]Research Institute for Molecular Pathology (IMP), Vienna BioCenter (VBC), Vienna, Austria. [3]Institute of Molecular Biotechnology (IMBA), Austrian Academy of Sciences, Vienna BioCenter (VBC), Vienna, Austria. [4]Gregor Mendel Institute of Molecular Plant Biology (GMI), Austrian Academy of Sciences, Vienna BioCenter (VBC), Vienna, Austria. [5]Center for Regenerative Medicine, Institute for Regeneration and Repair, University of Edinburgh, Edinburgh, UK. [6]Cancer Research UK Scotland Centre, University of Edinburgh, Edinburgh, UK. [7]Cell Signaling and Immunology, School of Life Sciences, University of Dundee, Dundee, UK. [8]Department for Clinical Neuroscience, NHS Lothian, Royal Infirmary Edinburgh, Edinburgh, UK. [9]Center for clinical Brain sciences, Institute for Neuroscience and Cardiovascular Research, University of Edinburgh, Edinburgh, UK. [10]These authors contributed equally: Pranvera Sadiku, Alejandro J. Brenes. ✉e-mail: abrenes@ed.ac.uk; karl.mechtler@imp.ac.at; sarah.walmsley@ed.ac.uk

**Table 1 | GBM patient demographics**

| Analysis type | Number of subjects (n) | Sex, n | Mean age (years ±S.D) |
|---|---|---|---|
| Flow cytometry | 15 | Male, 11 (73%) Female, 4 (27%) | 57 ± 11.2 |
| Mini-bulk and single cell proteomics | 6 | Male 4, (67%) Female 2, (33%) | 66 ± 8.9 |

firstly need to understand the relative functional contribution of these different neutrophil populations within the tumour site.

Alongside their conventional antimicrobial roles, neutrophils have emerged as important regulators of cancer. Their function remains paradoxical however, with both pro- and anti- tumourigenic effects reported at multiple stages of disease[4,5]. Neutrophils are short lived rapidly turned over cells, with the bone marrow producing more than 100 billion per day in homoeostasis[6]. They display significant transcriptional and functional heterogeneity in circulation, primary and metastatic tumour sites. This complexity is further amplified by cancer associated perturbations in myelopoiesis[5,7]. Single cell transcriptional analysis has been transformational in defining the presence of these different neutrophil states, and more recently has been used to predict a terminal transcriptional state in the tumour niche[8]. However, changes in rates of protein synthesis versus degradation[9] and the long-term storage of proteins within granules, together mean that transcriptomes cannot always predict effector functions. This is particularly relevant in both mature and immature neutrophil populations where a weak mRNA to protein correlation is observed[10]. Our objective, therefore, was to develop a more comprehensive proteomic platform to enable us to define neutrophil subsets by function using high resolution mass spectrometry (MS).

MS-based proteomics has proven to be a highly insightful tool to define neutrophil phenotypes in humans[10–14]. However, these bulk proteomic workflows required millions of cells for the analysis to be viable, limiting their use to peripheral blood neutrophils. Recent progress on sample processing[15–19], instrument sensitivity[20–22] and software tools[23,24], have facilitated the study of fewer cells[25,26], as well as enabling the proteomic characterisation of single cells. A new generation of mass spectrometers has significantly increased sensitivity and empowered single cell proteomics (SCP) to identify >5000 proteins per single HeLa cell[20,21]. This remarkable improvement has opened the possibility to study the functional heterogeneity of primary human cells composed of low protein content. It is important to note that neutrophils contain a much lower total protein mass, estimated to be less than 60 pg per cell, compared to HeLa cells, estimated to contain 250 pg of protein per cell.

Here we combine flow cytometry and MS-based proteomic workflows to study rare neutrophil populations, both from the peripheral blood and from glioblastoma (GBM) tumours. We performed a mini-bulk (500 cells) proteomic analysis of circulating mature CD10 + , immature CD10- low density neutrophils (LDN), normal density neutrophils (NDNs) as well as GBM tumour associated neutrophils (TAN), finding that TANs much more closely resemble mature CD10+ neutrophil populations, but with increased mitochondrial content. We performed ultra-sensitive SCP analysis of TANs to define functional neutrophil clusters within the GBM tumours. The power to identify >1100 proteins per single neutrophil provides us the ability to define the presence of discrete neutrophil functional clusters which we assign as armed, engaged, vital NETs, exhausted, lytic NETs, immunosuppressive and angiogenic and vascular immature. The added value of SCP has enabled the characterization of functional states not previously captured by single cell RNA sequencing (scRNAseq), as a result of imperfect correlation between the tissue neutrophil transcriptome and proteome. Consequently, our work stratifies neutrophil heterogeneity by effector function and opens new avenues to engage anti-tumoural neutrophil responses for GBM disease whilst also providing a platform for the study of neutrophils in disease at the single cell resolution.

## Results

To study rare neutrophil subsets in GBM, we developed mass spectrometry-based proteomics methods to work on low cell numbers. Similar to previous work[25], we established a mini-bulk method (500 cells) and a SCP method, both optimized for human neutrophils. Mini-bulk was used to analyze both peripheral blood neutrophils as well as GBM TANs, while only TANs were analyzed with SCP. GBM patient details are provided in Table 1. Both blood and tissue neutrophils were sorted by flow cytometry directly into 384 well plates containing the master mix (Supplementary Fig. 1A–C). Plates were stored at minus −80 °C, posteriorly processed on the cellenONE X1 and analyzed on an Orbitrap Astral equipped with a FAIMS Pro Duo interface using a 50 sample per day (SPD) method (Fig. 1A). This workflow enabled deeper coverage from just 500 cells and the pioneering single cell proteomic characterization of human neutrophils.

We compared the protein identifications of our two optimized low cell number methods to the standard bulk workflows which required 2 million neutrophils per sample (Fig. 1B). The data from all 3 proteomic methods were searched using Spectronaut 19, using stringent parameters optimized for mini-bulk and single cell, with a human SwissProt database that included isoforms and an immune cell specific contaminant database. Details on the parameters are provided in Table 2. As expected, the bulk proteomic workflow enabled the most comprehensive coverage of the neutrophil proteomes, with >5400 protein groups (proteins) identified per sample. However, in the bulk workflow a total 1.5 μg of peptides were injected into the mass spectrometer. The mini-bulk analysis enabled the identification of >3000 proteins per sample from just 500 cells, with the equivalent of -15 ng injected into the mass spectrometer. Finally, our SCP workflow enabled the identification of >1100 proteins per single neutrophil after tissue dissociation and cell sorting (Fig. 1C) from an estimated -30 pg of peptides injected. The mini-bulk and SCP workflows achieved remarkable coverage considering the low input.

Leveraging the bulk data, we calculated the estimated copy numbers using the proteomic ruler[27] (see methods) and studied proteins identified across all 3 workflows. The median estimated copy number of all proteins identified in bulk was -5500, thus 50% of proteins identified in bulk were present with fewer than 5500 estimated copies (Fig. 1D). Mini-bulk identified fewer low abundance proteins, with only 28% of the proteins identified in mini-bulk estimated to have <5500 copies based on the bulk data (Fig. 1D) and the median copy number of these proteins being -17,000 copies (Fig. 1E). In SCP only 20% of proteins identified were estimated to have <5500 copies based on the bulk data (Fig. 1D) and the median copy number being -34,000 copies (Fig. 1E). In summary, low abundance proteins were challenging to detect with low cell number workflows, however even SCP identified 20% of the proteins detected with <5500 copies in bulk.

Across all 3 methods, we were interested to examine the proteomic coverage of proteins that are vital to neutrophil function, focusing on neutrophil granules, metabolic proteins and proteins involved in immune signalling pathways. Our data showed that neutrophil granule proteins are highly abundant, with azurophilic granules (AG), specific granules (SG), ficolin granules (FG) and secretory vesicles (SV) all having a median copy number >60,000 copies. In bulk we could detect 682 granule proteins, of which 88% were also detected by mini-bulk and 80% by SCP (Fig. 1F), showing a remarkable capacity for SCP to characterize important neutrophil effector proteins. We next

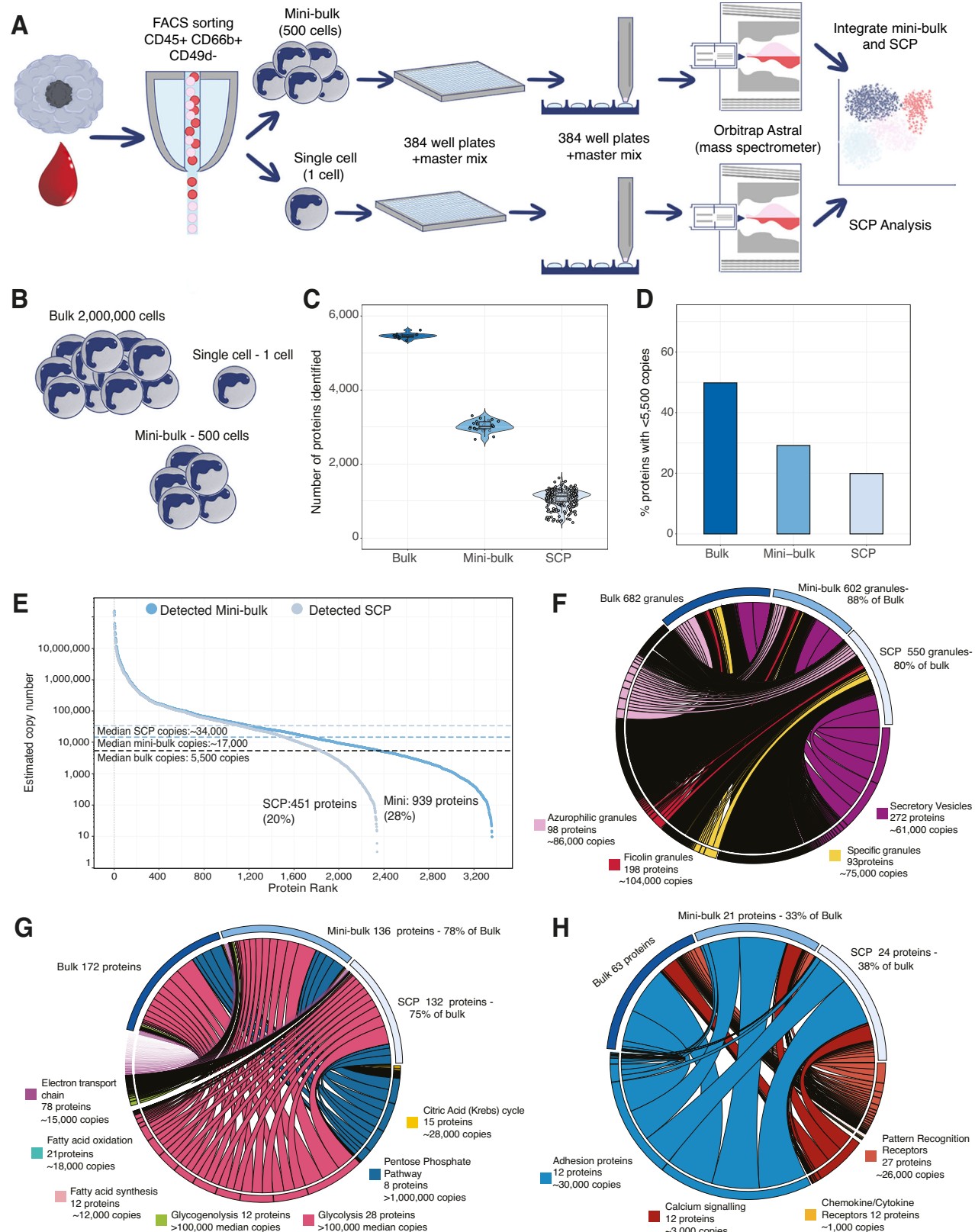

studied metabolic proteins which are vital to fuel neutrophil effector functions. The metabolic proteins were considerably less abundant than granule proteins but still displayed a median of >15,000 copies. 78% of the proteins identified by bulk were present in the mini-bulk analysis with excellent coverage (75% of proteins identified by bulk) also retained within the SCP data (Fig. 1G). We next focused on important neutrophil effector function regulators, such as adhesion proteins, proteins required for calcium signalling responses, cytokine and chemokine receptors and pattern recognition receptors (Fig. 1H). The adhesion proteins were highly abundant and showed good coverage, but low abundance chemokine receptors were difficult to detect even in bulk, and most were not detected with mini-bulk or SCP, showing the current limitations to identify important low abundance proteins.

**Fig. 1 | Low cell number mass spectrometry-based proteomic workflows.**
**A** Schematic showing the mini-bulk and single cell proteomic (SCP) experimental design. Neutrophils (CD45+, CD66b+, CD49d−) were sorted using flow cytometry directly into 384 well plates containing the master mix, later processed on the cellenONE X1 and analyzed on the Orbitrap Astral. **B** Schematic showing the number of cells that are analyzed in bulk, mini-bulk and single cell proteomics. **C** Violin and box plot showing the number of proteins identified in bulk ($n = 15$), mini-bulk ($n = 29$) and single cell proteomics ($n = 277$). **D** Percentage of proteins identified with less than 5500 copies (median copy number in bulk) across all 3 proteomic workflows. **E** Dot plot showing proteins identified in mini-bulk and SCP ordered by their median copy number in bulk. Chord diagrams showing the coverage of **F** granule proteins, **G** metabolic proteins and **H** immune signalling proteins across bulk, mini-bulk and single cell proteomics. For all chord diagrams the ribbons are sized by median copy number of each protein. For all boxplots, the top and bottom hinges represent the 1st and 3rd quartiles. The top whisker extends from the hinge to the largest value no further than 1.5× interquartile range (IQR) from the hinge; the bottom whisker extends from the hinge to the smallest value at most 1.5× IQR of the hinge. .

## Table 2 | Spectronaut parameters

| Parameter | Value |
|---|---|
| Precursor Qvalue cutoff | 0.01 |
| Protein Qvalue cutoff (experiment) | 0.01 |
| Protein Qvalue cutoff (run) | 0.01 |
| Precursor PEP cutoff | 0.15 |
| Protein PEP cutoff | 0.15 |
| Protein LFQ method | Quant2.0 |
| Quantity MS method (bulk) | MS2 |
| Quantity MS method (mini-bulk & single cell) | MS1 |
| Cross-run normalisation | Off |
| Major (protein) grouping | Protein Group ID |
| Major group quantity | Peptide sum |
| Major group top N | Off |
| Minor (peptide) grouping | By Stripped Sequence |
| Minor group quantity | Median precursor quantity |
| Minor group top N | Off |
| Enzymes/cleave rules | Trypsin/P |
| Missed cleavages | 2 |

## Mini-bulk enables the analysis of mature and immature blood neutrophils and TANs

To define GBM neutrophil subsets by cell surface marker expression, we first undertook flow cytometry analysis of peripheral blood from healthy control (HC) and GBM patients. The evaluated markers were CD10, CD66b and CD11b. CD10 is also known as common acute lymphoblastic leukaemia antigen and has previously been identified as a marker of neutrophil maturity[28]. CD66b, also known as carcinoembryonic antigen-related cell adhesion molecule 8, and CD11b, known as integrin alpha M, are proteins vital for cell adhesion and migration and are widely used as neutrophil activation markers[29–32]. GBM patients have been shown to have a significantly elevated leukocyte count[3] and altered proportions of lymphocytes and neutrophils, with an increased neutrophil to lymphocyte ratio in the peripheral blood compared to healthy controls[33]. Expansion of a circulating low density neutrophil (LDN) subset composed of a mixture of mature and immature cells has previously been reported in lung and breast cancer patients with distinct phenotypic properties[5]. We detected the presence of an immature CD10- neutrophil subset in the blood of GBM patients which was significantly increased in the LDN fraction (Supplementary Fig. 2A–C). Furthermore, this immature subset had features of activation with high CD66b and reduced CD11b expression (Supplementary Fig. 2D, E).

Given the identification of multiple GBM neutrophil subsets by our flow cytometry analysis, we decided to further characterize these populations using our mini-bulk method (Supplementary Data 1). We stratified the neutrophils by density and maturity, with CD10+ representing mature neutrophils and CD10- immature neutrophils. Blood LDN CD10-, blood LDN CD10+, blood normal density neutrophils (NDN) CD10+, and CD45+CD66b+CD49d− TANs derived from 6 GBM

patients and blood NDN CD10+ neutrophils from 6 healthy control donors were analyzed (Fig. 2A). Across all 5 neutrophil populations it was possible to identify >3000 proteins per 500 cell sample (Fig. 2B) with a median of >14,500 peptides (Fig. 2C), showing the robustness of the method. Relative intensity based absolute quantification (riBAQ) was used to normalize and analyze the mini-bulk data (see methods). An initial principal component analysis (PCA) of the riBAQ data displayed unexpected results. The neutrophil populations displayed no clear separation based on density, with LDN and NDN CD10+ groups showing limited separation (Fig. 2D). TANs also occupied the same dimensional space as the CD10+ section and it was only the CD10− population that showed a clear segregation. Studying the top 25 proteins that influenced the PCA revealed ribosomal proteins and PCNA marking this separation (Supplementary Fig. 3A).

We performed a direct comparison between CD10+ LDN and NDN and found there were no proteins significantly changed between the two populations (Fig. 2E), with all granule subsets showing no difference between mature LDN and NDN (Supplementary Fig. 3B–E), suggesting density is not the primary axis of proteomic variation in neutrophils, and raising the possibility that lipid content might contribute to the changes in cell density. We next compared mature CD10+ NDN and LDN to immature LDN CD10−. We found 29% of proteins significantly different between the NDN CD10+ and LDN CD10− populations (Fig. 2F), with very similar results seen when comparing LDN CD10+ and LDN CD10- (Supplementary Fig. 3F). As most mature neutrophils exist in the normal density state, we focused on the NDN CD10+ comparison and found a total of 526 proteins were significantly higher in abundance in the LDN CD10- immature neutrophils. A gene ontology (GO) analysis of these proteins revealed they were enriched in translational, DNA, and mitochondrial related GO terms (Fig. 2G). The immature neutrophils displayed >330% increased abundance of ribosomal proteins compared to mature LDN and NDN (Fig. 2H, I), suggesting higher translational capacity. Mitochondrial metabolism proteins were also increased, with the electron transport chain >165% higher in immature cells compared to mature neutrophils (Fig. 2J), with similar increases in DNA replication proteins (Supplementary Fig. 3G–M).

There were 355 proteins that were significantly more abundant in mature NDNs compared to immature LDNs. A GO analysis highlighted these were enriched in processes related to cell membrane, migration, and inflammatory response (Fig. 2K), suggesting the potential for superior migratory and effector capacity. A higher abundance of proteins reported to associate with increased neutrophil maturation status was observed, including core components of the NADPH oxidase, granule proteins MMP8, MMP9, S100A8, S100A9 (Fig. 2L) and the cell surface markers CD11b, CD18, CD16b (Fig. 2M)[29,34–36]. Interestingly, although the glucose transporter SLC2A3 is not a known marker of neutrophil maturation, we find its abundance increases in line with changes in abundance of neutrophil maturation surface markers.

## TANs resemble mature circulating neutrophil populations

We were interested to understand whether tumour associated neutrophil (TAN) populations had a unique proteomic signature. TANs displayed higher coefficient of variation (CV) when compared to the

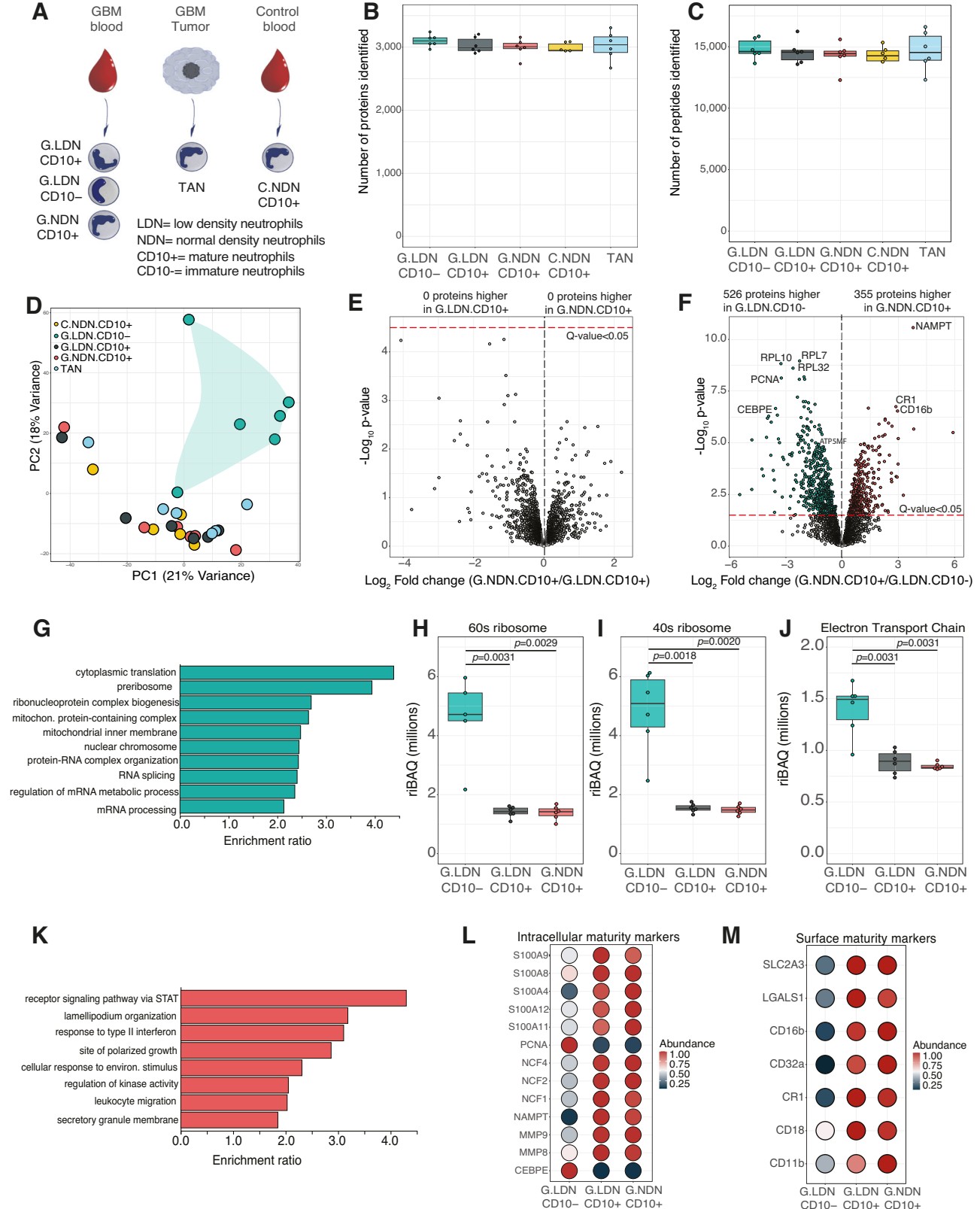

blood derived populations (Fig. 3A). In line with the PCA overview, which displayed no clear separation of TANs from CD10+ populations, a direct comparison of TANs to mature CD10+ NDN reflected only 8 proteins significantly changed in abundance (Fig. 3B), suggesting limited differences compared to normal density mature neutrophils. Of the few proteins changed, TANs displayed increased abundance of brain specific markers such as myelin basic protein (MBP) and monoamine

oxidase B (MAOB) while L-selectin (SELL), the cell adhesion molecule important for trans-endothelial migration, was higher in the NDN CD10+. We then performed a direct comparison of TANs to LDN CD10-, and found this revealed 553 proteins changed in abundance (Fig. 3C). Proteins with higher abundance in TANs were enriched for GO terms relating to migration and adhesion (Fig. 3D) and demonstrated increased abundance of maturity markers (Supplementary Fig. 4A–H).

**Fig. 2 | Mature (CD10+) and immature (CD10−) neutrophil proteomes.**
**A** Schematic showing the experimental design for mini-bulk proteomic analysis of human circulating blood and tumour associated neutrophil (TAN) populations including: glioblastoma patient derived blood low density mature neutrophils (G.LDN.CD10+); glioblastoma patient derived blood low density immature neutrophils (G.LDN.CD10−), glioblastoma patient derived blood normal density mature neutrophils (G.NDN.CD10+); healthy control derived blood normal density mature neutrophils (C.NDN.CD10+); and glioblastoma TANs. Box plots showing the (**B**) number of proteins identified in the mini-bulk analysis (C.NDN.CD10+ $n = 5$, for all other conditions $n = 6$) and the (**C**) number of peptides identified in the mini-bulk analysis (C.NDN.CD10+ $n = 5$, all others $n = 6$). **D** Principal component analysis (PCA) of the mini-bulk proteomic samples ($n = 29$). Volcano plots comparing (**E**) G.NDN.CD10+ to G.LDN.CD10+ and (**F**) G.NDN.CD10+ to G.LDN.CD10−. **G** Gene ontology (GO) enrichment analysis for proteins significantly increased in abundance in G.LDN.CD10−. Boxplots ($n = 6$ across all conditions) showing the summed protein abundance of **H** 60S ribosomal proteins, **I** 40S ribosomal proteins, **J** the electron transport chain proteins across G.LDN.CD10-, G.LDN.CD10+, G.NDN.CD10+. **K** GO enrichment analysis for proteins significantly increased in abundance in G.NDN.CD10+. Dot plot showing **L** intracellular maturity markers and **M** cell surface maturity markers across G.LDN.CD10−, G.LDN.CD10+, G.NDN.CD10+. For all boxplots, the top and bottom hinges represent the 1st and 3rd quartiles. The top whisker extends from the hinge to the largest value no further than 1.5× interquartile range (IQR) from the hinge; the bottom whisker extends from the hinge to the smallest value at most 1.5× IQR of the hinge. All volcano plots show the $p$-values and fold changes. For panels (**E**, **F**, **H**–**J**) all $p$-values were calculated with limma using Empirical Bayes statistics for differential expression and are two-sided. All points above the red line have a $q$-value < 0.05. For all GO plots, all GO terms shown have an FDR < 0.05.

Proteins decreased in abundance in TANs related to ribosomal, and DNA replication proteins (Fig. 3E–G). Markers of proliferation were also significantly lower in TANs compared to the immature cells (Supplementary Fig. 4I, J). Together these highlight that TANs are much more similar to CD10+ mature neutrophil populations than the immature CD10− neutrophils. Interestingly however, TANs did display an intermediate mitochondrial phenotype, when compared to CD10− LDN and CD10 + NDN (Fig. 2H, I), suggesting the potential for mitochondrial metabolic adaptation within the tumour niche.

## Single cell proteomics identifies distinct neutrophil states in Glioblastoma

To investigate neutrophil heterogeneity in GBM, we utilized ultra-sensitive MS-based single cell proteomics (SCP). We optimized a workflow where GBM TAN were FACS sorted into 384 well plates and analyzed on the Orbitrap Astral with FAIMS Pro interface using a 50 sample per day workflow (see Methods). For this analysis the mini-bulk proteomic data was integrated to deconvolute the SCP data set, (Fig. 1A) with immature populations within the SCP data identified using the maturity markers defined in the NDN CD10+ to LDN CD10− mini-bulk comparison.

A total of 330 neutrophils derived from tumours of the same 6 patients used in the mini-bulk analysis were processed. As previously mentioned, the MS-based raw files were searched with Spectronaut v19.4, using software parameters with increased stringency compared to the default settings (Table 2). From these results, cells with less than 400 proteins identified were filtered out (see methods), leaving a total of 277 cells for further analysis (Supplementary Data 2, Supplementary Fig. 5A–F). The intensity data produced by Spectronaut was normalized using a relative iBAQ (riBAQ) like approach (see methods). riBAQ is an intensity measure which normalizes for protein size. Proteins with higher mass and longer amino acid sequences frequently have more theoretical tryptic peptides (peptides produced by trypsin digestion) than proteins with smaller mass. riBAQ divides the protein intensity by the total number of potential tryptic peptides, thus normalizing for differences in protein size. Hence, riBAQ provides a measure of intensity that better correlates to absolute protein abundance and is more comparable across different experiment types[37]. The riBAQ-like data was analyzed in Seurat with batches integrated using harmony (see methods). Across the 277 cells a median of >1100 proteins per single neutrophil were identified (Fig. 1B). The preliminary analysis of the riBAQ like intensity data across the 2000 most variable protein features (see methods) in Seurat led to the identification of seven distinct neutrophil populations (Fig. 4A), which we assign as (1) armed, (2) engaged, (3) vital NETs, (4) exhausted, (5) immunosuppressive and angiogenic, (6) lytic NETs and (7) vascular immature clusters. All clusters contained data from at least three patients, though the exhausted cluster was dominated by two patients. The most abundant population was the armed neutrophil subset, and the least abundant the vascular immature subset (Fig. 4B).

## Vascular signature of lytic NETosis and immature GBM populations with reduced frequency of immature neutrophils in the tumour niche

To assess the contribution of immature neutrophils to the tumour neutrophil pool we integrated the protein level maturity markers identified in the mini-bulk data with the SCP and identified one population, representing 8% of the cells, displaying an immature signature (Fig. 4C). This population, labelled as vascular immature, was defined by increased abundance of PCNA and CEBPE (Fig. 4C) with concurrent increased abundance of mitochondrial metabolic enzymes including PDHB, IDH3A and ACAT1, mitochondrial membrane proteins including TOMM5 and IMMT (Fig. 4D), and overall ribosomal proteins including RPL4, RPL7 and RPL15 (Fig. 4E). The marker of cell proliferation PCNA and transcription factor CEBPE are expressed in immature cells and involved in DNA replication and repair and regulation of granule protein expression, respectively[38,39]. In keeping with the immature phenotype, mitochondrial metabolism has been shown to be reduced during neutrophil differentiation[40]. This combined signature closely resembled the LDN CD10- population identified by our mini-bulk analysis. This immature tumour neutrophil population was reduced in frequency compared to the CD10- peripheral blood compartment (33%) identified by flow cytometry (Supplementary Fig. 2C). Within this immature tumour population we also identified a signature that suggested it was located within the vasculature, with elevated levels of complement and coagulation proteins (Fig. 4F).

A second neutrophil population was also observed to display high abundance of complement proteins, fibrinogens, F2 and F9 coagulation factors, clusterin and plasminogen (Fig. 4F). This population displayed loss of highly abundant proteins that are released during lytic NETosis. Complement activation has previously been shown to stimulate NET formation and NETs themselves can activate the complement cascade and also serve as a scaffold for coagulation factors[41]. This is in keeping with recent findings by Adrover et al. reporting the presence of a neutrophil population driving vascular occlusion through NET formation and aggregation with platelets at sites of fibrin deposition[42]. The lytic NETosing neutrophil cluster displayed an 80% reduction in the abundance of primary azurophilic granules (Fig. 4G), 83% reduction in total histone proteins (Fig. 4H) and an 85% reduction in nuclear membrane proteins (Fig. 4I). Similarly pronounced reductions were observed in secondary granules, ficolin granules and secretory vesicles (Supplementary Fig. 6A–C). In keeping with a cell undergoing lytic NETosis these neutrophils had also lost their protein capacity to perform effector functions or respond to extra-cellular signals[43] with a sharp reduction in metabolic enzymes required to fuel neutrophil effector functions (Supplementary Fig. 6D) and cell membrane proteins required to respond to DAMPs and PAMPs (Supplementary Fig. 6E).

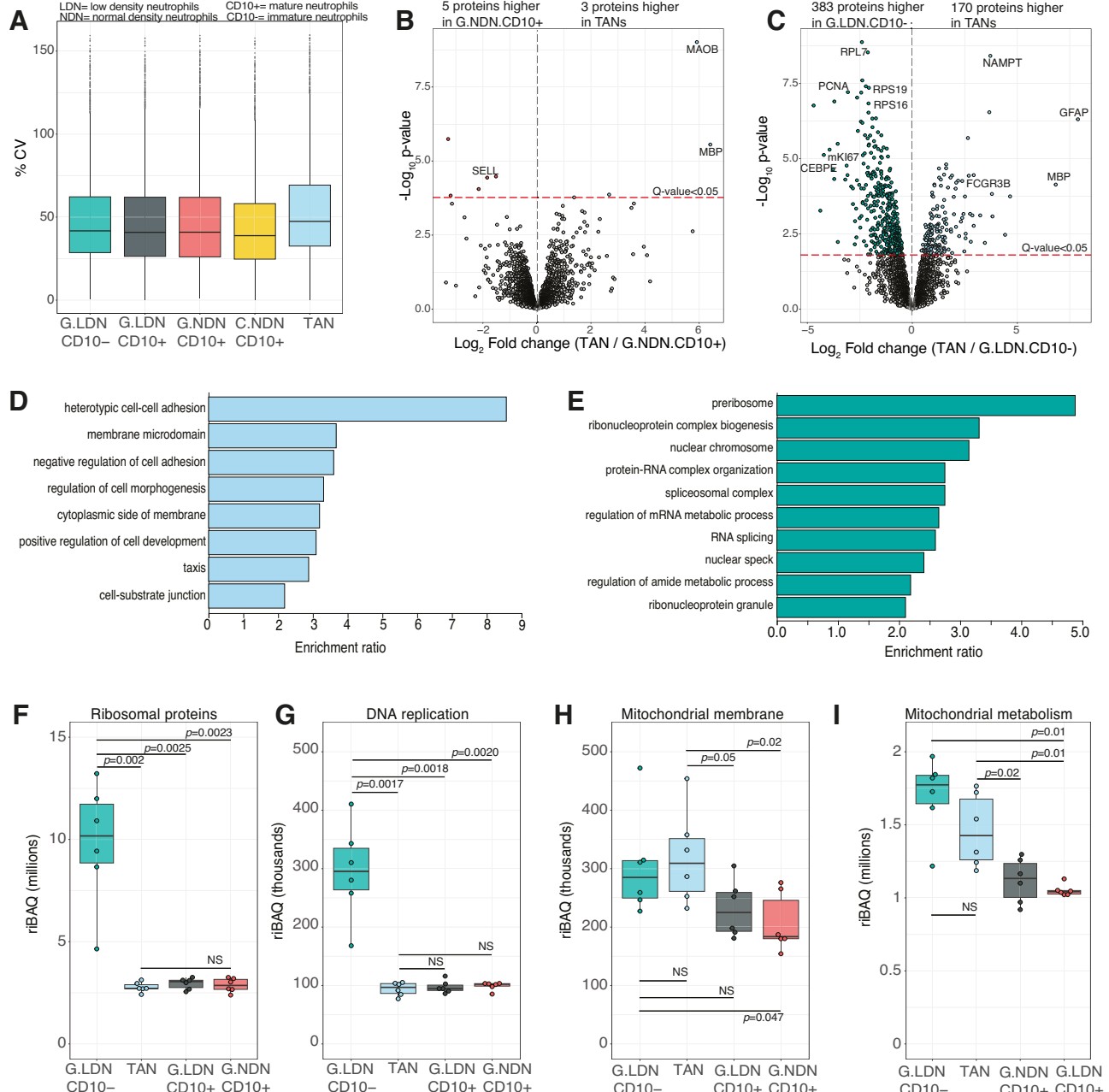

**Fig. 3 | TANs resemble mature neutrophils with a mitochondrial phenotype.**
**A** Boxplot showing the percentage coefficient of variation (CV) for all mini-bulk populations including: glioblastoma patient derived blood low density mature neutrophils (G.LDN.CD10+); glioblastoma patient derived blood low density immature neutrophils (G.LDN.CD10−), glioblastoma patient derived blood normal density mature neutrophils (G.NDN.CD10+); healthy control derived blood normal density mature neutrophils (C.NDN.CD10+); and glioblastoma patient derived tumour associated neutrophils (TAN). Volcano plots comparing **B** G.NDN.CD10+ to TANs and **C** G.LDN.CD10- to TANs. Gene ontology (GO) enrichment analysis for proteins **D** significantly increased in abundance in TANs compared to G.LDN.CD10- and **E** significantly decreased in abundance in TANs compared to G.LDN.CD10-.

Boxplots (n = 6 across all conditions) showing the summed riBAQ for all **F** ribosomal proteins, **G** DNA replication proteins, **H** mitochondrial membrane proteins and **I** mitochondrial metabolism proteins. For all boxplots, the top and bottom hinges represent the 1st and 3rd quartiles. The top whisker extends from the hinge to the largest value no further than 1.5× interquartile range (IQR) from the hinge; the bottom whisker extends from the hinge to the smallest value at most 1.5× IQR of the hinge. All volcano plots show the p-values and fold changes. For panels (**B**, **C**) all p-values were calculated with limma using Empirical Bayes statistics for differential expression and were two-sided. All points above the red line have a q-value < 0.05. For panels (**F**–**I**) all protein family p-values were calculated using a two-sided Welch's t test. For all GO plots, all GO terms shown have an FDR <0.05.

---

## Neutrophils degranulate and undergo vital NETing within the GBM tumour

Of the mature neutrophil populations present within the tumour, SCP identified cell states that correlated with a functional trajectory. The most abundant population, the armed neutrophils, has a signature consistent with recent extravasation[44] with the highest abundance of cytoskeletal and motility related proteins (Fig. 5A) including

ITGB2 (CD18) and ITGAM (Cd11b), vital integrins forming the MAC1 complex. They also display high histone abundance (Fig. 5B) and the highest abundance of granule proteins[3,45] (Fig. 5C). Engaged neutrophils demonstrate signs of granule release, exemplified by reductions in myeloperoxidase (MPO) and calprotectin subunit S100A8 content, and a trajectory towards vital NETing (Fig. 5D, E) where loss of granule proteins and secretory vesicles[46] (Fig. 5F) is paired with a

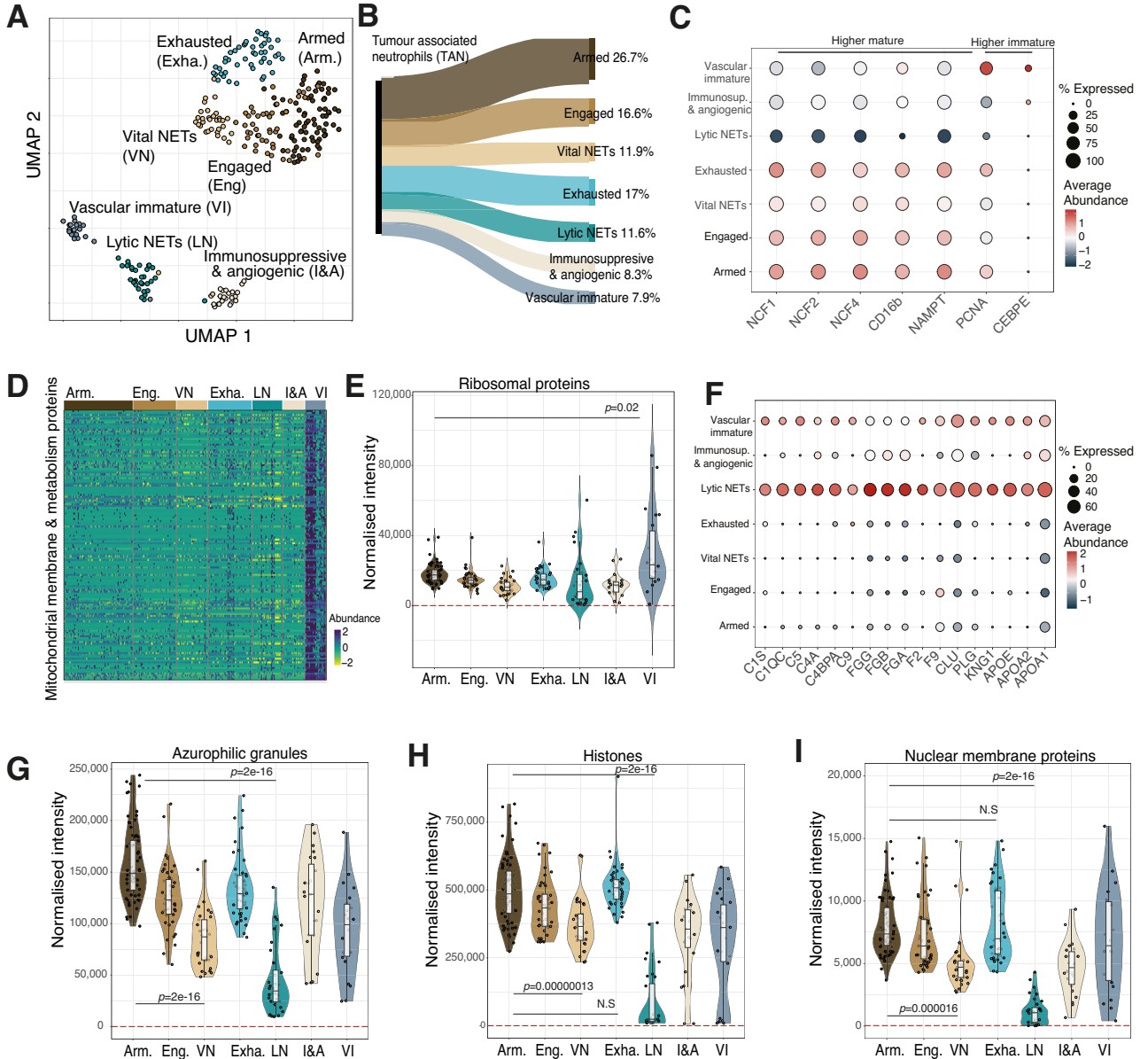

**Fig. 4 | Single cell proteomic analysis of TANs identified discrete functional clusters. A** Uniform manifold approximation and projection (UMAP) showing the tumour associated neutrophil (TAN) clusters. (*n* = 277). **B** Sankey diagram showing the proportions represented by each cluster. **C** Dotplot showing maturity marker proteins across all clusters. **D** Heatmap showing mitochondrial metabolism and mitochondrial membrane proteins across all cells ordered by cluster. **E** Boxplot showing the summed normalised intensity of all ribosomal proteins. **F** Dotplot showing complement and coagulation proteins across all clusters. Boxplots showing the summed normalised intensity of all **G** azurophilic granules, **H** histones

and **I** nuclear membrane proteins across all clusters. Armed (Arm. *n* = 76), Engaged (Eng. *n* = 46), Vital NETs (VN, *n* = 33), Exhausted (Exh. *n* = 47), Lytic NETs (LN, *n* = 32), Immunosuppressive and angiogenic (I&A, *n* = 23) and Vascular Immature (VI, *n* = 22). For all boxplots the top and bottom hinges represent the 1st and 3rd quartiles. The top whisker extends from the hinge to the largest value no further than 1.5× interquartile range (IQR) from the hinge; the bottom whisker extends from the hinge to the smallest value at most 1.5× IQR of the hinge. For panels (**G**–**I**) all protein family boxplots *p*-values were calculated using a two-sided Welchs *T* test.

reduction in histones[47] (Fig. 5G, H) and nuclear and cell membrane proteins (Fig. 5I).

A discrete mature neutrophil state is also observed in which reduced expression of granule proteins and secretory vesicles is associated with retained levels of histone proteins but with metabolic anergy. We have termed these exhausted neutrophils. These cells have a significant reduction in proteins involved in vital metabolic processes including glycolysis, glycogenolysis and the pentose phosphate pathway (Fig. 5J), exemplified by low level expression of the key regulatory proteins glucose transporter GLUT 3[48] (SLC2A3) and glycogen

phosphorylase[49,50] (PYGL) (Fig. 5K, L). In keeping with a reduced capacity to respond, this dysfunction also extends to regulators of neutrophil function including the NADPH oxidase regulator RAC2 (Fig. 5M), and the calcium binding protein calmodulin (Fig. 5N). Interestingly, this population was also defined by high abundance of GABBR1, a GABA receptor associated with innate lymphoid cell inhibition[51] (Fig. 5O) and CD2BP2, a CD2 binding protein with unknown neutrophil effector functions (Fig. 5P). Together, this data suggests that neutrophils arrive armed and poised to respond, they get engaged with potential end points of exhaustion or vital NETing (Fig. 5Q).

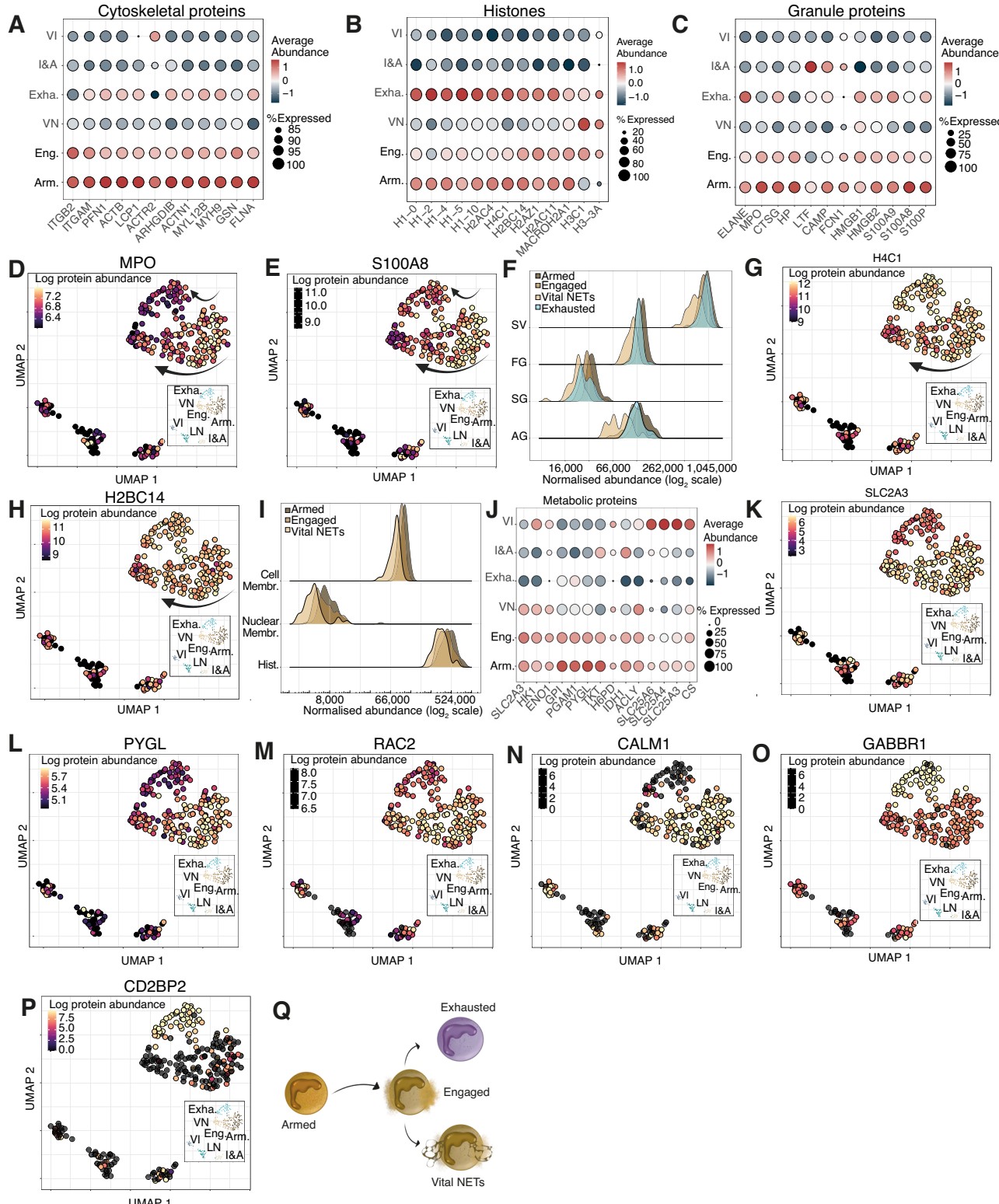

**Fig. 5 | Predicted trajectory between neutrophil functional states.** Dotplots for **A** cytoskeletal proteins, **B** histones and **C** granule proteins across armed (Arm.), engaged (Eng.), vital NETs (VN), exhausted (Exh.), immunosuppressive and angiogenic (I&A) and vascular immature neutrophils (VI). Uniform manifold approximation and projection (UMAPs) for **D** myeloperoxidase (MPO) and **E** S100 calcium-binding protein A8 (S100A8). **F** Ridgeplot for azurophilic granules (AG), specific granules (SG), ficolin granules (FG) and secretory vesicles (SV) across armed, engaged, vital NETs and exhausted clusters. UMAPs for **G** H4 Clustered Histone 1 (H4C1) and **H** H2B Clustered Histone 14 (H2BC14). **I** Ridgeplot of histones (Hist.),

nuclear membrane proteins (Nuclear Membr.) and cell membrane proteins (Cell Membr.) across armed, engaged and vital NETs clusters. **J** Dotplot showing metabolism related proteins. UMAPs for **K** Solute Carrier Family 2 Member 3, GLUT3 (SLC2A3), **L** Glycogen Phosphorylase Liver form (PYGL), **M** Rac Family Small GTPase 2 (RAC2), **N** Calmodulin 1 (CALM1), **O** Gamma-Aminobutyric Acid Type B Receptor Subunit 1 (GABBR1) and **P** CD2 Cytoplasmic Tail Binding Protein 2 (CD2BP2). (**Q**) Schematic showing the proposed neutrophil trajectory. All Ridgeplots show the summed normalised abundance of specific protein families. For all UMAPs (*n* = 277) the colour scale shows the log normalised protein abundance.

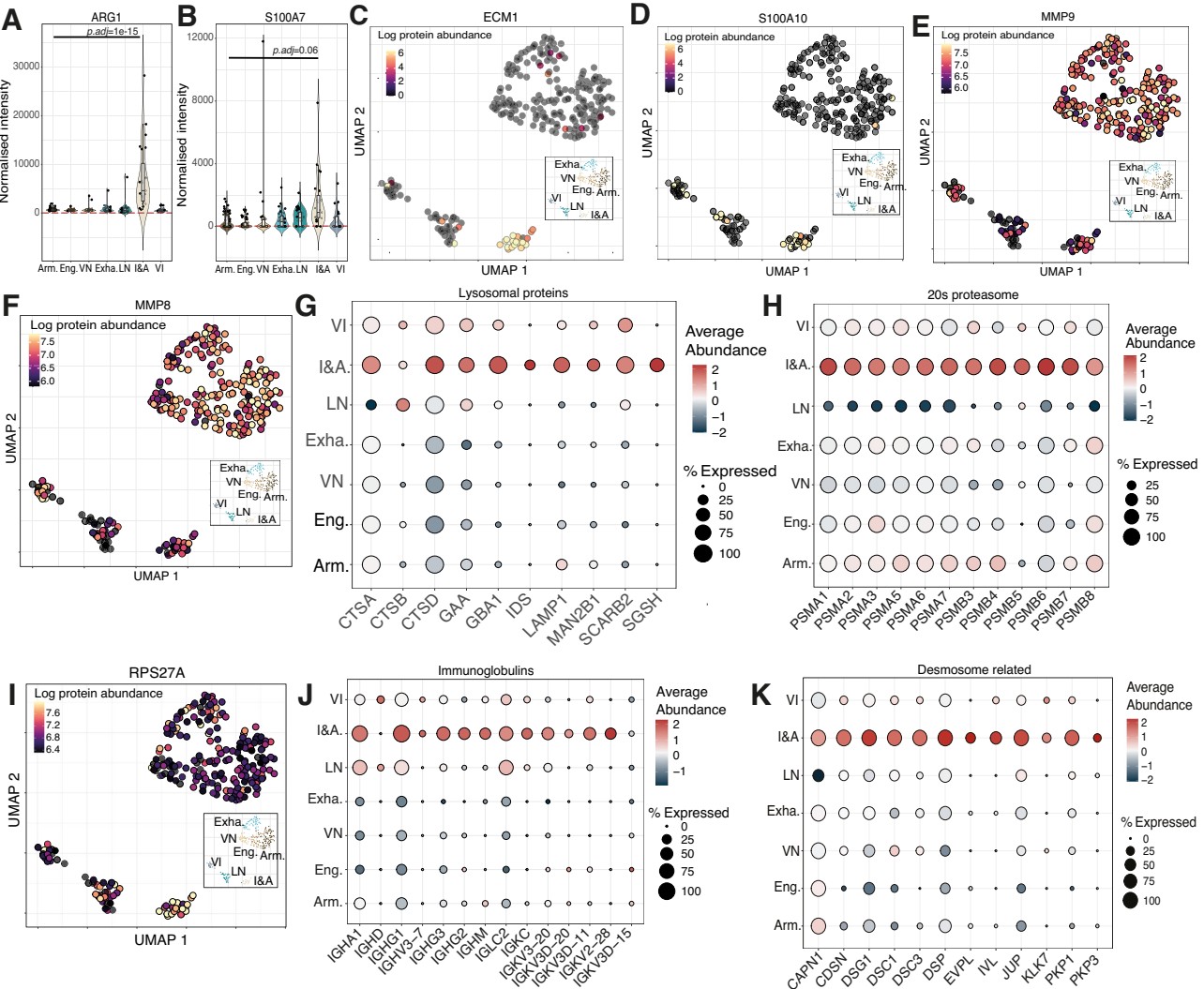

**Fig. 6 | Presence of an immunosuppressive and angiogenic neutrophil subset.** Boxplots for **A** Arginase 1 (ARG1) and **B** S100 Calcium Binding Protein A7 (S100A7) showing the normalized intensity across all clusters. Uniform manifold approximation and projection (UMAPs) for **C** Extracellular Matrix Protein 1 (ECM1), **D** S100 Calcium Binding Protein A10 (S100A10), **E** Matrix Metallopeptidase 9 (MMP9) and **F** Matrix Metallopeptidase 8 (MMP8). Dotplots showing **G** core lysosomal proteins and the **H** 20S proteasome proteins. **I** UMAP for the ubiquitin protein (RPS27A). Dotplot showing **J** immunoglobulin proteins and **K** Desmosome related proteins. For all UMAPs (*n* = 277) the colour scale shows the log normalized protein abundance. Armed (*n* = 76), Engaged (*n* = 46), Vital NETs (*n* = 33), Exhausted (*n* = 47), Lytic NETs (*n* = 32), Immunosuppressive and angiogenic (*n* = 23) and Vascular Immature (*n* = 22). For all boxplots the top and bottom hinges represent the 1st and 3rd quartiles. The top whisker extends from the hinge to the largest value no further than 1.5× interquartile range (IQR) from the hinge; the bottom whisker extends from the hinge to the smallest value at most 1.5× IQR of the hinge. For panels (**A**, **B**) the individual protein boxplot adjusted *p*-values were calculated using Seurat with logistic regression and are two-sided.

## Immunosuppressive & angiogenic neutrophils have a phagocytic signature

The proteomic data also uncovered the presence of a distinct neutrophil state within the tumour that displayed an immunosuppressive signature[52], with high abundance of Arginase 1 (ARG1) and S100A7 (Fig. 6A, B). This cluster was further defined by an increase in abundance of the pro-angiogenic factors ECM1 (extracellular matrix protein 1) (Fig. 6C) and S100A10 (Fig. 6D) thus we have termed these immunosuppressive and angiogenic (I&A) neutrophils. A reduction in the matrix metalloproteinases, MMP9 (Fig. 6E) & MMP8 (Fig. 6F) was also revealed. This is of interest as neutrophil release of MMPs has also been shown to promote angiogenesis through the degradation of the extracellular matrix[53]. In addition, immunosuppressive and angiogenic neutrophils displayed evidence of enhanced lysosomal capacity with increased abundance of core lysosomal proteins (Fig. 6G), including Cathepsin D (CTSD), lysosomal associated membrane protein 1 (LAMP1) and scavenger receptor class B member 2 (SCARB2). This was associated with a global increase in protein degradation machinery, including the core components of 20S ubiquitin-proteasome complex (Fig. 6H), with evidence of enhanced activity provided by higher levels of free ubiquitin (Fig. 6I). In keeping with enhanced phagocytic activity, this cluster also demonstrated an increase in abundance of extracellular immunoglobulin (Ig) proteins (Fig. 6J) and non-neutrophil specific proteins normally related to the desmosomes (Fig. 6K). Within the tumour niche, we have therefore identified a subset of pro-tumoural neutrophils that can phagocytose and process extracellular proteins via the lysosomal compartment, activate T cell suppression pathways and promote angiogenesis.

## Discussion

Single cell RNAseq has been instrumental in helping to define neutrophil transcriptional subsets, however due to the strong discordance

between mRNA and protein abundance[10], an understanding of neutrophil functional heterogeneity in health and disease is lacking. Although the transcriptome can be used to predict cellular state, it is the proteome that is the key driver of biological function especially in post-mitotic cells such as neutrophils. The bulk proteomes of immune cells including neutrophils have over the years been extensively studied, mapping cellular protein signatures and function to understand disease mechanisms and identify biomarkers. However, bulk proteomics obscures the single cell resolution necessary to capture complex biological tissue cell-state heterogeneity. Important advances in single cell proteomics including the original isobaric labelling methods[54,55], nanodroplet sample processing techniques[15,17,56], label free instrument optimisations[57] and even spatial solutions[58,59] have enabled the quantification of thousands of proteins from a single cell. Despite these developments, the lack of amplification strategies for proteins has mostly limited single cell proteomic (SCP) analysis to larger cells containing a higher protein content such as blastomeres, oocytes and HeLa cells[60]. Whilst a single HeLa cell is estimated to contain between 150 and 250 pg of protein[61], a single human neutrophil is estimated to contain between 30 and 60 pg of protein, posing a substantial challenge for SCP analysis[60,61].

To facilitate profiling of neutrophil proteomes with low protein input, we have developed a proteomics workflow optimised for low cell numbers, capable of measuring from a 500 neutrophil mini-bulk down to a single FACS sorted human neutrophil dissociated from glioblastoma tumours. Our high sensitivity workflow has enabled the detection of >3000 proteins from 500 neutrophils and importantly >1100 proteins per single human TAN, with good coverage of the highly abundant neutrophil effector proteins exemplified by the granule proteins (80% SCP coverage), metabolic proteins (75% SCP coverage), and reasonable coverage of immune signalling proteins (38% SCP coverage). This workflow has allowed us to generate a pioneering human single cell tissue neutrophil proteomic resource.

Neutrophils are highly dynamic cells with the capacity to adapt to the tissue niche. There is compelling evidence to show that in addition to re-programming within the tumour micro-environment, newly formed neutrophils and their progenitors can also be perturbed in the bone marrow long before they reach the tumour site[5,62]. We find that GBM alters circulating neutrophils with an expansion of a low density immature neutrophil population with a distinct proteomic signature. Our mini-bulk proteomic survey of 500 blood CD10− low density neutrophils (LDN), CD10+ LDN, CD10+ normal density neutrophils (NDN) and CD45+CD66b+CD49d− tumour associated neutrophils (TANs) using ultra-sensitive mass spectrometry revealed proteomic similarities between CD10+ blood neutrophils and TANs, however it also highlighted clear differences between the CD10− LDN and TAN populations. This is of interest given the significant contribution of LDN to the circulating neutrophil pool and work detailing deterministic reprogramming of both immature and mature neutrophils by the tumour niche[8]. Single cell proteomic survey of neutrophils harvested from resected glioblastoma tumours at time of surgery subsequently identified the presence of an immature neutrophil cluster based on the abundance of specific maturity markers defined by mini-bulk as well as mitochondrial and ribosomal protein content. However, this accounted for only 8% of the tumour neutrophil populations and was associated with an intra-vascular signature of complement and coagulation proteins. We therefore propose that in human GBM CD10− immature neutrophils are retained within the vasculature. It remains an open question as to whether there is either active exclusion of immature neutrophils with anti-tumoural capacity by prevention of their vascular egress or whether there is rapid transition of these cells into a terminally differentiated pro-tumoural state.

Importantly, our study also reveals the concurrent existence of neutrophil states with the potential capacity to engage either pro- (immunosuppressive, lytic NETotic) or anti- (armed, engaged) tumoural responses. Our work also highlights the value and contribution of SCP, as multiple neutrophil states uncovered by this data would be invisible to RNAseq. Specifically, NETotic neutrophils were characterized by prominent reductions in the abundance of histones and granule proteins whilst degranulating neutrophils were characterized by a reduction in granule protein content with conserved histone abundance. As granule proteins are preserved within mature neutrophils long after their corresponding transcripts are no longer present[10], these functional clusters are only exposed when proteomic analysis is performed. SCP also enables characterization of less frequent populations, for example we were able to identify a cluster of neutrophils undergoing lytic NETosis not visible within the mini-bulk data. The lytic NETosis population displayed increased abundance of complement and coagulation proteins, giving insights into their predicted vascular localisation and mechanism of activation. This is of interest given the recent description via 3D imaging in murine cancer models of tumour-elicited vascular-restricted neutrophils that promote tissue necrosis via NETosis and can be blocked to reduce metastatic spread[42]. The proteomic signature of GBM neutrophils undergoing lytic NETosis complements this description of compartment restricted neutrophil effector functions raising the important potential for targeted therapeutic intervention to limit the tissue necrosis associated with poor survival outcomes in GBM[63]. It also highlights the value of proteomics to study phagocytic cells capable of pinocytosis, as we have previously shown neutrophils can uptake and use extracellular proteins, the presence of which can work as proxy markers for their microenvironment[11,64]. In this case it is unclear if the complement and coagulation proteins are on the cell surface or internalized, however their detection still provides valuable biological insights.

Our study provides a platform to explore human neutrophil functional heterogeneity with single cell resolution and should be considered a starting point for the investigation of neutrophil contribution to homoeostasis and disease pathogenesis. Future work will be required to understand what determines the different neutrophil states in cancers such as GBM and whether they are a consequence of exposure to local cues within the tumour niche e.g. hypoxia, specific cell-cell interactions, or a consequence of pre-determined trajectories that are manifest upon exposure to the tumour micro-environment. Further developments in the field of single cell spatial proteomics will allow us to start to address these important questions. For example, can the peri-vascular niche play a critical role in compartmentalization of these different effector responses as previously reported for GBM macrophage interactions[65]. Does neutrophil phagocytosis and clearance of cell debris and tumour antigens in the necrotic core itself engage immunosuppressive pro-angiogenic responses? How do neutrophil populations evolve during tumour progression, treatment and recurrence?

In summary (Fig. 7), single cell proteomics now unlocks the door to defining functional changes in neutrophil subsets that are subverted in disease and amenable to therapeutic targeting. Moreover, the SCP and mini-bulk workflows presented here have the potential to be widely applicable to human leucocytes in any tissue as well as disease setting, further enhancing scientific development in the field of immune cell biology in context of cancer and beyond.

## Methods

### Human participants
The collection of peripheral blood from healthy male and female control participants was approved by the Centre for Inflammation Research Blood Resource Management Committee (AMREC #15-HV-013) and recruited from the University of Edinburgh Centre for Inflammation Research Blood Resource. Exclusion criteria for healthy controls included the following: infection with any blood borne diseases, previous or current intravenous drug abuse, anaemia, blood

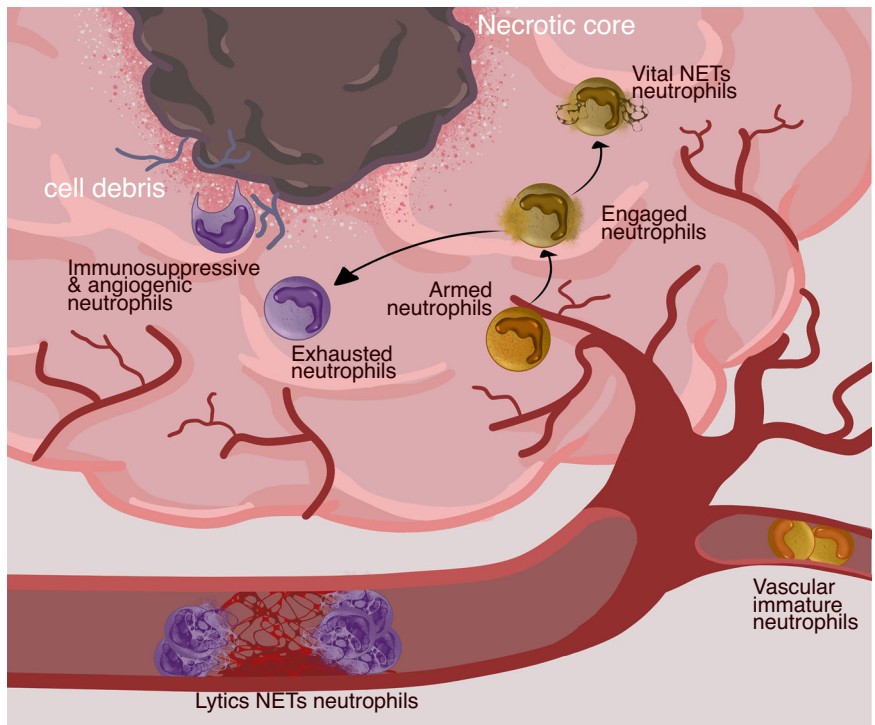

**Fig. 7 | Neutrophil functional state overview.** Schematic showing an overview of the proposed neutrophil functional states in human glioblastoma characterised by our single cell proteomic workflow where >1100 proteins were identified per tumour associated neutrophil. Discrete functional clusters are assigned as armed, engaged, vital neutrophil extracellular traps (NETs), exhausted, lytic NETs, immunosuppressive and angiogenic, and vascular immature.

clotting disorders, anticoagulant drug therapy, regular use of steroids and/or under the age of 16 years old. The collection of peripheral blood and tumour tissue from glioblastoma (GBM) male or female patients was approved by the Lothian NRS Bioresource, (East of Scotland Research Ethics Committee REC 1 #13/ES/0126) and recruited between January 2015 to 2025 from NHS Lothian hospitals, Edinburgh, Scotland, UK. Informed consent was obtained from all participants prior to sample collection. The study was approved by AMREC and the East of Scotland Research Ethics Committee REC and was conducted in accordance with the Declaration of Helsinki.

### Isolation of human blood neutrophils

For the isolation of normal density neutrophils (NDN) from healthy control whole blood and NDN and low density neutrophils (LDN) from whole blood of GBM patients, immunomagnetic isolation of neutrophils directly from whole blood using the EasySep™ Direct Human Neutrophil Isolation Kit (StemCell Technologies) was performed as per the manufacturer's instructions, followed by a discontinuous Percoll™ gradient as previously described[66]. Briefly, the immunomagnetically isolated neutrophil pellet was resuspended in 49% Percoll™, carefully layered above 61% Percoll™ already layered onto 73% Percoll™ and centrifuging at $720 \times g$ for 20 min at 20 °C, with brake off and acceleration set at 1. NDN were harvested from the 61%/73% interface and LDN were harvested from the 49%/61% interface followed by washing with DPBS without $Ca^{2+}$ or $Mg^{2+}$ (Gibco) prior to phenotyping or sorting.

### Processing of GBM tumour tissue

Fresh human tumour tissue collected during surgery was dissected using a scalpel and exposed to enzymatic digestion by incubation in complete RPMI medium supplemented with 10% v/v HI-FBS (PAN-Biotech), 1% v/v penicillin-streptomycin (1000 U/L, Gibco), DNase (1000 U/L, Roche Diagnostics) and Collagenase Type IV (0.1% w/v, ThermoFisher Scientific) in sterile conditions at 37 °C in a humified

atmosphere containing 5% $CO_2$ for 30 min. The tissue digest was mixed intermittently every 10 min during the 30 min incubation. The tissue digest was then passed through a 100 µm Fisherbrand™ cell strainer into a 50 mL polystyrene Falcon™ tube and this process was repeated using 70 µm and then 40 µm Fisherbrand™ cell strainers. The volume of cell suspension was adjusted to 50 mL with 0.9% saline and centrifuged at $350 \times g$ for 6 min at 20 °C (acceleration 9, deceleration 5) for cell pelleting. Hypotonic red cell lysis was then performed by first re-suspending the cell pellet in 10 mL of 0.2% w/v NaCl in sterile water followed by the addition of 10 mL 1.6% w/v NaCl and 0.1% w/v D-(+)-glucose in sterile water and centrifuged at $350 \times g$ for 6 min at 20 °C (acceleration 9, deceleration 5). The cell pellet was re-suspended in 5 mL 0.9% saline and used for flow cytometry phenotyping or fluorescence-activated cell sorting (FACS) of tumour neutrophils.

### Fluorescence-activated cell sorting (FACS) of human GBM tumour neutrophils

Human GBM tumour tissue was dissected and digested to a single cell suspension as described above. For samples required for SCP and mini-bulk proteomics, cells were first incubated with 2 ml of 1:200 diluted FC Block for 10 min at 4 °C, followed by staining in 2 ml of an antibody cocktail containing BV421-anti-CD45 (1:200 final dilution), APC-anti-CD66b (1:57 final dilution), PE-anti-CD49d (1:200 final dilution) and PE-Cy7-anti-CD10 (1:57 final dilution) for 30 min at 4 °C. Appropriate unstained and fluorescence minus one (FMO) controls were also generated. Following three washes with PBS w/o $Ca^{2+}$ and $Mg^{2+}$, 500 cells of healthy control circulating NDN CD10+ and circulating GBM normal density (NDN) CD10+, low density (LDN) CD10+ mature, immature LDN CD10-, tumour associated neutrophils (TANs) were sorted for mini-bulk proteomic analysis. 66 single GBM TANs from each patient were sorted for single cell proteomic (SCP) analyses. A FACS Aria Fusion sorter *(Becton Dickinson)* was used to collect cells in 384 well plates containing 1 µL of a cell lysis master mix per well (See mini-bulk and single cell proteomics sample preparation).

## Human GBM neutrophil phenotyping by flow cytometry

GBM NDN and LDN ($0.1 \times 10^6$/test) isolated from Percoll gradients and digested human GBM tumour tissue were stained with an antibody cocktail mix containing FITC-anti-CD66b (1:40), PE-Cy7-anti-CD11b (1:40), PerCP-Cy5.5-anti-CD49d (1:40), APC-Cy7-anti-CD10 (1:40) for 30 min at 4 °C with appropriate unstained and FMO controls. Cells were washed and re-suspended in FACS buffer and acquired using BD LSRFortessa™ flow cytometer (Beckton Dickinson), with compensation performed using BD FACSDiva™ software version 8.0 and data analyzed in FlowJo version 10.2.

## Bulk proteomics sample preparation

2 million cell pellets were lysed in 100 μL lysis buffer (5% SDS, 10 mM TCEP, 50 mM TEAB) and were then shaken at room temperature for 5 min at 1000 rpm, followed by boiling at 95 °C for 5 min at 500 rpm. Samples were then shaken again at RT for 5 min at 1000 rpm before being sonicated for 15 cycles of 30 s on/ 30 s off with a BioRuptor (Diagenode). Benzonase was added to each sample and incubated at 37 °C for 15 min to digest DNA. Samples were then alkylated with the addition of iodoacetamide to a final concentration of 20 mM and incubated for 1 h in the dark at 22 °C. Protein concentration was determined using EZQ protein quantitation kit (Invitrogen) as per manufacturer instructions. Protein digestion was performed using S-TRAP micro columns (Protifi). Proteins were digested with trypsin at 1:10 ratio (enzyme:protein) for 2 h at 47 °C. Digested peptides were eluted from S-TRAP columns using 50 mM ed peptides were dried overnight before being resuspended in 40 μL 1% formic acid ready for analysis by data independent acquisition (DIA) mass spectrometry.

## Bulk proteomics mass spectrometry

The bulk proteomics data was acquired in DIA mode on an Orbitrap Exploris 480 (Thermo Scientific) coupled with an UltiMate 3000 RSLC nano (Thermo Scientific™). 1.5 μg of peptide per sample were injected. Two buffers were used: buffer A (0.1% formic acid in Milli-Q water (v/v)) and buffer B (80% acetonitrile and 0.1% formic acid in Milli-Q water (v/v). Samples were loaded at 10 μL/min onto a trap column (100 μm × 2 cm, PepMap nanoViper C18 column, 5 μm, 100 Å, Thermo Scientific™) equilibrated in 0.1% trifluoroacetic acid (TFA). The trap column was washed for 3 min at the same flow rate with 0.1% TFA then switched in-line with a Thermo Scientific™, resolving C18 column (75 μm × 50 cm, PepMap RSLC C18 column, 2 μm, 100 Å). Peptides were eluted from the column at a constant flow rate of 300 nL/min with a linear gradient from 3% buffer B to 6% buffer B in 5 min, then from 6% buffer B to 35% buffer B in 115 min, and finally to 80% buffer B within 7 min. The column was then washed with 80% buffer B for 4 min and re-equilibrated in 3% buffer B for 15 min. Two blanks (1% formic acid buffer) were run between each sample to reduce carry-over. The column was kept at a constant temperature of 50 °C.

The data was acquired using an easy spray source operated in positive mode with spray voltage at 2.445 kV, and the ion transfer tube temperature at 250 °C. The MS was operated in DIA mode. A scan cycle comprised a full MS scan (m/z range from 350 to 1650), with RF lens at 40%, AGC target set to custom, normalised AGC target at 300%, maximum injection time mode set to custom, maximum injection time at 20 ms, microscan set to 1 and source fragmentation disabled. MS survey scan was followed by MS/MS DIA scan events using the following parameters: multiplex ions set to false, collision energy mode set to stepped, collision energy type set to normalized, HCD collision energies set to 25.5, 27 and 30%, orbitrap resolution 30,000, first mass 200, RF lens 40%, AGC target set to custom, normalized AGC target 3000%, microscan set to 1 and maximum injection time 55 ms. Data for both MS scan and MS/MS DIA scan events were acquired in profile mode.

## Mini-bulk and single cell proteomics sample preparation

Single neutrophils and mini-bulk samples (500 cells) were sorted into fresh 384 well plates (Thermo Scientific™ Armadillo PCR Plate, 384-well, #12657516) each well containing 1 μL of master mix. For 0 cell samples, no cells were sorted into the respective master mix-containing wells. The master mix contained 0.2% n-dodecyl-ß-D-maltoside (DDM, #D4641-500MG, Sigma Aldrich, Germany), 100 mM triethylammonium bicarbonate (TEAB, #17902-500 ML, Fluka Analytical, Switzerland), and 3 ng/μL trypsin (Trypsin Gold, #V5280, Promega, USA) in ultra-pure water. Directly after sorting, samples were stored at −80 °C. Just prior to lysis and digestion, plates were thawed and centrifuged for 3 min at 300 × g at 4 °C. Lysis and digestion was performed similarly to earlier works[16]. In brief, plates were placed in the CellenONE X1 (Cellenion, Lyon, France, #F00C) at 50 °C and 85% humidity to limit evaporation. This is followed by incubation for 2 h at 50 °C at 85% relative humidity inside the instrument. Samples were kept hydrated every 15 min by automated addition of 500 nL ultra-pure water to each well. After 30 min of incubation, an additional 500 nL of 3 ng/μL trypsin were added which replaces one hydration step. After lysis and digestion, 2.5 μL of 0.1% trifluoro acetic acid (TFA, Thermo Fisher Scientific, #28903) with 5% dimethyl sulfoxide (Avantor, #83673.230) were added to the respective wells for quenching and storage. Plates were then stored at −20 °C before LC-MS/MS measurement.

## Mini-bulk and single cell proteomics mass spectrometry

All samples were analyzed using a Vanquish Neo UHPLC (Thermo Fisher Scientific, #VN-S10-A-01) operated in trap-and-elute mode and coupled to an Orbitrap Astral mass spectrometer (Thermo Fisher Scientific, #BRE725600) equipped with a FAIMS Pro Duo interface (ThermoFisher Scientific, #OPTON-20068). Analyte separation was performed at 50 °C using a 25 cm × 75 μm C18 UHPLC packed emitter column (Ion Opticks Pty Ltd, # AUR3-25075C18-TS, Collingwood, Australia) connected to the mass spectrometer via an EASY-Spray™ ion source (Thermo Fisher Scientific, ES081). Before separation, peptides were trapped on a 5 mm PepMap™ Neo trap cartridge (Thermo Fisher Scientific, #174500) using a flow of 10 μL/min of 0.1% TFA. Loading was done using combined control of flow (10 μL/min) and pressure (max. 800 bar). Loading volume was set to automatic. Samples were cooled to 7 °C in the autosampler and covered with a silicone mat. 5 μL were injected per sample to ensure complete aspiration of the entire sample volume using 0.2 μL/s draw speed and 2 s draw delay with bottom detection on. For separation, 0.1% formic acid in LC-MS/MS grade water (Thermo Fisher Scientific, # 10188164) as solvent A and 0.08% formic acid in 80% acetonitrile (Thermo Fisher Scientific, #10118464) as solvent B were used. After sample loading, peptides were eluted using the following gradient as previously described[20] to achieve a throughput of roughly 50 samples per day: 0.0 min 450 nL/min 1%B − 0.1 min 450 nL/min 4%B − 1.9 min 450 nL/min 12%B − 2.0 min 200 nL/min 12%B − 12.0 min 200 nL/min 22.5%B − 19.5 min 200 nL/min 40%B − WASH PHASE − 22.0 min 300 nL/min 99%B − 25.0 min 300 nL/min 99%B.

An electrospray voltage of 1.9 kV was applied for peptide ionization with a static carrier gas flow of 3.5 L/min. The ion transfer tube temperature was set to 280 °C, expected peak width to 6 s, the default charge state to 2 and advanced peak determination was enabled. Total method length was set to 27 min. MS1 were recorded in positive polarity in the orbitrap at 240,000 resolution for a scan range of 400−800 m/z. The AGC target was set to 500% (=5E6 absolute AGC target) with 100 ms as maximum injection time. A single FAIMS CV of −48 V was used and profile chosen as data type. RF lens (%) was kept at 45.

Fragmentation spectra were recorded using DIA with 20 m/z isolation windows without any overlap and window placement

optimization on. DIA MS2 spectra were recorded in the Astral mass analyzer for a precursor range of 400–800 m/z and a scan range of 150–2000 m/z. Collision energy was set to 25% NCE and the AGC target to 800% (=80,000 absolute AGC target). The maximum injection time was set to 40 ms for mini-bulk and 80 ms for single and 0 cell samples. Centroid was chosen as data type and 0.6 s selected as loop control. DIA window type was set to Auto and mode to m/z range.

## Number of samples analyzed

For the bulk proteomics a total of 15 samples were analyzed. For mini-bulk a total of 30 biological samples were analyzed. For single cell proteomics a total of 342 samples were analyzed, 12 of which represented the 0-cell master mix only quality controls.

## Spectronaut analysis

The bulk and mini-bulk were searched with Spectronaut[67] 19.7, the single cell was searched with Spectronaut 19.4. For all 3 searches the default parameters were altered to increase stringency similar to Baker et al.[68]. The details are specified in Table 1. All searches were performed using directDIA against a human SwissProt + isoforms database (November 2024) and an immune cell specific contaminant fasta file. The fasta files are included in the PRIDE submissions. No variable modifications were included in the searches.

## Copy number calculation

For all data in Fig. 1, estimated protein copy numbers were calculated using the healthy control participants from the bulk data. Copy numbers were calculated from the mass spectrometry-based intensity data using the proteomic ruler[27]. The median copy numbers for bulk, mini-bulk and SCP were calculated using the bulk estimations but for proteins identified with each methodology.

## Proteomic data normalisation

For mini-bulk the intensity based absolute quantification (iBAQ) was used and divided by the median intensity across each sample to generate the relative iBAQ (riBAQ) for all proteins across all samples. For SCP, due to issues in the iBAQ calculation with Spectronaut 19.4, an iBAQ like measure was calculated. In brief, the protein intensity was divided by the protein molecular weight and multiplied by the median molecular weight of all detected proteins (47,485 Da). By using the median molecular weight instead of the sum of all molecular weights, the intensity scale is kept in the same order of magnitude as the original data. Across each sample this molecular weight corrected intensity was divided by the sample median, after filtering out all 0 values, producing a riBAQ like quantitation.

## Coefficient of variation (cv) calculations

The CVs were calculated using the proteomicsCV[69] package v0.4.0. The protein intensities were fed to the package after performing the previously stated riBAQ normalization. All proteins were used in this calculation and no filtering for outliers was performed.

## Bulk and mini-bulk differential expression analysis

The differential expression analysis was performed in R using the Bioconductor package limma[70] v3.60.6 using lmFit with the method='robust' parameter and eBayes with the 'robust=TRUE' parameters. Q values were calculated using the Bioconductor package qvalue v2.36.

## Mini-bulk volcano plot filters

For all mini-bulk volcano plots only proteins that were detected in at least 2 independent replicates, in both conditions being compared, were considered. Any proteins detected in 1 replicate or less in any conditions were filtered out.

## Significant proteins

For the mini-bulk analysis, proteins were considered to be significantly changed in abundance if their $q$-value < 0.05. For single cell proteomics they were considered significant if the Bonferroni adjusted $p$-values were <0.05.

## Mini-bulk gene ontology (GO) analysis

The GO analysis was performed using WebGestalt[71] with a minimum number of analytes per category set to 8, Significance Level set to FDR 0.05, and weighted set cover, affinity propagation and k-Mediods enabled. The functional databases were set to Gene Ontology Biological Process and Cellular Component both non-redundant. The background was set to all proteins detected in at least two replicates in both NDN CD10+ and LDN CD10-, and two replicates in the TAN and LDN CD10-.

## Principal component analysis

All mini-bulk based PCAs were analyzed in R using only proteins that were identified in all samples and after log10 transformation using the prcomp function part of the stats v4.4.1 package.

## Mini-bulk sample filtering

One control sample had 10-fold lower number of proteins and peptides identified and was a clear outlier. This sample was filtered from the posterior analysis.

## Single cell proteomics cell filtering

The 0 cell runs were used as QC for filtering. A threshold of >1.75-fold higher than the maximum number of identifications in the 0 cell runs (Supplementary Fig. 5A) was set and rounded up to 400 proteins. A total of 277 cells were analyzed after filtering.

## Seurat pipeline

The single cell proteomics data was analyzed using Seurat[72] v.5.2.1. The intensity data were normalised as described above and this normalisation was used in Seurat. The min.features were set to 400 (filtering cells with less than 400 proteins identified) and FindVariableFeatures used nfeatures = 2000. Total number of principal components (npcs) was set to 35. The data were batch corrected using the R package harmony v.1.2.3 and grouping the data by patient. FindNeighbours() used the harmony input with a total of 35 dimensions, FindClusters() used a resolution of 1.45. The number of clusters were determined using the ElbowPlot() method and set to 7.

## Single cell population markers

For the population markers the FindMarkers() function in Seurat was used. A fold-change threshold of 0.25, logistic regression as the selected method, and a Bonferroni corrected $p$-value < 0.05 was considered significant.

## Single cell protein family significance testing

All protein family significance were calculated using the sum of the intensity of all proteins within the family and calculated using Welchs T test in R. A $p$-value < 0.05 was considered significant

## Single cell proteomics data visualisations

The doplots and UMAP were generated with Seurat. The sankey diagram was generated with Sankeymatic. The violin plots were generated in R using ggplot2 v3.5.1.

## Reporting summary

Further information on research design is available in the Nature Portfolio Reporting Summary linked to this article.

## Data availability

All raw proteomics data and search engine results generated in this study has been uploaded to the PRoteomics IDEntifications Database (PRIDE)[73], part of the ProteomeXchange[74] consortium, and has been made open access. The bulk proteomics data is available under accession PXD061859, the mini-bulk proteomics data is available under accession PXD061052 and the single cell proteomic data under accession PXD061065. The processed single cell proteomic data is available within the Immunological Proteome Resource[75]. The protein intensities and differential expression data are provided in the Supplementary Information/Supplementary Data files. Source data are provided with this paper.

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

## Acknowledgements

Flow cytometry data were generated with support from the QMRI and IRR Flow Cytometry and Cell Sorting Facility, within the University of Edinburgh. This work was funded by a CRUK Cancer Immunology pro-ject award C62207 (S.R.W.), Wellcome Trust Senior Clinical Fellowship awards 098516, 209220 (S.R.W.), Discovery science award 225778 (S.R.W.) and a Wellcome Trust Clinical Research Career Development Fellowship award 224637/Z/21/Z22 (E.W.). This work was also supported by the infrastructure funding fourth call 2022/01 of the Austrian Research Promotion Agency (K.M.) and the project LS20-079 of the Vienna Science and Technology Fund (K.M.), the P35045-B project (grant DOI 10.55776/P35045) and the project PAT4142423 (Grant DOI 10.55776) of the Austrian Science Fund (K.M.). We thank the Protein Chemistry Facility and acknowledge the VBCF for instrument access. We also thank the Lothian NRS bioresource, the consultant neuro-anaesthetists and registrars of the department of clinical neurosciences, NHS Lothian for assistance in recruiting, consenting and obtaining samples from patients with Glioblastoma. For the purpose of open access, the author has applied a CC BY public copyright license to any Author Accepted manuscript version from this submission.

## Author contributions

Conceptualization: S.R.W.; Methodology: P.S., A.J.B., R.L.M., L.R., S.M.P., K.M.; Investigation: P.S., A.J.B., L.R., P.C., G.V.S., A.Z., M.A.S.-G., E.R.W., I.L., A.J.M.H., I.A., A.B., A.M., S.R., G.M.M., H.M., C.M.B., S.J., R.G., F.A.M., P.M.B.; Funding acquisition: S.R.W., K.M.; Supervision: P.S., K.M., S.R.W.; Writing – original draft: P.S., A.J.B., S.R.W.; Writing – review & editing: P.S., A.J.B., R.L.M., G.V.S., K.M., S.R.W.

## Competing interests

S.M.P. is a founder, consultant and shareholder of Trogenix Ltd, a bio-tech that is working on advanced therapies for cancers of unmet need. He is an inventor on patents owned by the University of Edinburgh that

have been licensed to Trogenix. The remaining authors declare no competing interests.
