## [Transparent Peer Review file · Nature Communications]

Single cell proteomic analysis defines discrete neutrophil functional states in human glioblastoma

Corresponding Author: Professor Sarah Walmsley

Version 0:

Reviewer comments:

Reviewer #1

(Remarks to the Author)

Sadiku, Brenes, and coworkers reported an exciting study using a highly sensitive proteomics workflow to investigate neutrophil populations in the context of human glioblastoma. In the past several years, Neutrophil was known to be present in many solid tumors and autoimmune diseases. However, due to its post-mitotic nature, it is largely missed in single-cell RNA-seq and spatial transcriptomics studies. Their roles in these diseases are often overlooked due to the limited technologies available (FACS, bulk-scale measurement, or RNA). The authors directly addressed this gap by coupling the most advanced proteomics technology. The study reveals a clearer picture of Neutrophil populations and functions, including well-known armed and NETosis-associated populations. Importantly, they also observed a pro-tumoural population existed in GBM, which could help to generate a potential therapeutic intervention approach. The manuscript is well written, and the data quality is high. It is an excellent example of how single-cell proteomics can advance our understanding of the diverse functions of immune cells in tumor microenvironments.

I have a couple of comments for the authors to consider:

1. One of the main challenges to applying omics technologies in a clinical setting is the heterogeneous expression pattern among different patients. In this study (mini-bulk and single cell), the authors indicated that the cells were collected from 6 GBM patients. The reviewer is wondering if the author observed patient-specific signatures (clusters).
2. The authors employed riBAQ to perform single-cell data processing (normalization). There is not enough information to explain why such a method is used. Single-cell proteomics is a relatively new field, and this study represents a first-of-its-kind application in a clinical setting. More information will help the field to adopt this method in other disease studies.
3. The data missingness is another well-known challenge in single-cell proteomics. How is the data missingness handled in this study?
4. Similarly, most of the volcano plots are plotted with P-value with variable cutoff, likely due to the use of multiple hypothesis testing. To avoid confusion, it is better to plot using the same way (adjusted P value with fixed cutoff).
5. It is interesting to observe that the lytic NETosis Neutrophils have highly abundant complement and coagulation proteins (blood signatures). Since lytic NETosis Neutrophils lose most of their cytoplasmic and nuclear context in the tumor, it is not clear how these blood proteins get inside of the Neutrophils after NETosis.
6. Check Figure 6G. The labels are quite distinct from those of other figures.
7. Ref. 28 is missing

Reviewer #2

(Remarks to the Author)

Summary statement

Here, using a small cohort of glioblastoma patients and healthy controls, the authors perform an in-depth characterization of circulating and tumor associated neutrophils (TANs) via bulk, mini-bulk, and single cell proteomic analyses. They clearly outline the sensitivity of each approach and demonstrate quantification of key functional neutrophil proteins down to the single cell level. Flow cytometry sorting based on cell surface marker expression reveals an expanded population of circulating immature, CD10- low density neutrophils (LDNs) in glioblastoma patients. These immature CD10- neutrophils are molecularly distinct from mature CD10+ neutrophil subsets and TANs, which molecularly resemble each other, supporting that TANs are likely mature neutrophils. Integration of the mini-bulk data with 277 single TAN proteomes identified seven neutrophil subtypes, functionally classified by differential protein expression. Single cell analysis confirms that TANs are predominantly mature, with a small proportion of immature TANs suggested to be vascular associated. The six mature TAN subtypes exhibit somewhat distinct proteomic signatures linked to cell states, defined by specific protein markers, enabling observable functional trajectories for TAN subtypes. These analyses highlight TAN turnover and posit an ideal molecular signature that could be utilized for therapeutic intervention (engaged). The authors also identify an interesting TAN subtype suggested to be immunosuppressive, warranting further investigation in the context of tumors.

Overall, this manuscript presents a compelling rationale for proteome level characterization of neutrophils in glioblastoma patients. It is a valuable reference and offers a framework to investigate single cell proteomes, even in small cells with low protein input. The integration of single-cell and bulk data is particularly innovative, highlighting the complexity of bulk datasets, incorporating single cell sensitivity measurements, and enabling functional characterization of cell populations. The authors additionally perform an in-depth analysis that allows the proposal of a clinically relevant cell state mechanism that could serve as a model for future single cell studies. Minor comments and clarifications are listed below to enhance the manuscript. One point of curiosity concerns the classification of resected tumors, given changes in WHO glioblastoma criteria during the patient recruitment period. However, these comments do not diminish the overall enthusiasm of the manuscript.

Remarks

1. Additional detail or a diagram related to CD markers and their functional meanings would be useful to provide in the main text or to extended data Figure 2 for those outside of the field.
 2. Why is median and not total summed molecular weight of all detected proteins used for normalization?
 3. Regarding extended figure 3, is it interesting that the 25 protein differentiators of all neutrophil subtypes are not differentially expressed between LDN (CD10+) and NDN (CD10+)? Any other thoughts on difference between mature neutrophils with differing densities?
 4. Sample stratification and any additional relevant covariates across individual human subjects or sex should be made clear within the figures and/or methods, especially as it relates to the mini-bulk and single cell datasets.
 5. In 2021, during the time period of cohort recruitment, the definition of glioblastoma changed to a very specific classification based on genetic testing and histological classification (PMID: 34185076). Is there any additional depth to tumor classification that can be achieved in these samples? Do the authors know if a patient's tumor misclassification would impact the results? In Figure 2D it looks like there might be a single patient that is an outlier.
 6. Given the frequency of males to females diagnosed with glioblastomas are there any sex differences observed in the neutrophil populations?
 7. Where is the comparison for CD10+ LDN to CD10- LDN and why is that not the comparison for mature (CD10+ NDN) vs immature (CD10- LDN) neutrophils that only differ by their CD10 expression and not additionally density?
 8. Why are LDN CD10+ neutrophils excluded from extended data figure 3 and 4. Is this much variation in detected CD10 expected for CD10+ neutrophils?
 9. Appreciate the Spectronaut parameters table in the methods.
 10. It would be helpful to have specific references that add validity to maturation markers identified from the proteomic mini-bulk comparison of LDN CD10- and NDN CD10+ referred to in lines 213-215. These markers are used frequently to classify single cell TAN populations in this manuscript, therefore it would be helpful to know exactly how they were selected. The authors should be more upfront if they are referring to maturity and immaturity markers as basically differentially expressed proteins in CD10- comparisons. The markers in Figure 2K, 2L and 4C do not match.
 11. The utilization of proteins to classify functional neutrophil subtypes is clever. However, a substantial lack of references for the purported functional relationship of each marker protein throughout the results are missing in the manuscript.
- Minor comments
12. There is a space needed in line 203
 13. For GO enrichment analysis of Figure 2, what is the threshold for significance.

Reviewer #3

(Remarks to the Author)

Version 1:

Reviewer comments:

Reviewer #1

(Remarks to the Author)

I read through the response letter and find the authors have addressed all of my comments. I support the publication of the manuscript

Reviewer #2

(Remarks to the Author)

Thank you for taking the time to read and address the comments that I and the other reviewers have made to this impressive manuscript. I believe that this is ready to progress to the next stage of the publication process. Congratulations to this team for such an impressive body of work that I will undoubtedly cite in the future and feel grateful to have seen in the present form.

Reviewer #3

(Remarks to the Author)

We were delighted by the overall positive feedback provided by the reviewers, including reference to data quality, innovative data integration and generation of a framework in which to study single cell proteomes. Please find below a point-by-point response to each of the reviewer comments.

Reviewer #1 (Remarks to the Author):

Sadiku, Brenes, and coworkers reported an exciting study using a highly sensitive proteomics workflow to investigate neutrophil populations in the context of human glioblastoma. In the past several years, Neutrophil was known to be present in many solid tumors and autoimmune diseases. However, due to its post-mitotic nature, it is largely missed in single-cell RNA-seq and spatial transcriptomics studies. Their roles in these diseases are often overlooked due to the limited technologies available (FACS, bulk-scale measurement, or RNA). The authors directly addressed this gap by coupling the most advanced proteomics technology. The study reveals a clearer picture of Neutrophil populations and functions, including well-known armed and NETosis-associated populations. Importantly, they also observed a pro-tumoural population existed in GBM, which could help to generate a potential therapeutic intervention approach. The manuscript is well written, and the data quality is high. It is an excellent example of how single-cell proteomics can advance our understanding of the diverse functions of immune cells in tumor microenvironments.

We are delighted that the reviewer finds our work of high quality and that they have highlighted how they see this work contributing understanding to the diverse function of neutrophils in the tumour microenvironment.

I have a couple of comments for the authors to consider:

1. One of the main challenges to applying omics technologies in a clinical setting is the heterogeneous expression pattern among different patients. In this study (mini-bulk and single cell), the authors indicated that the cells were collected from 6 GBM patients. The reviewer is wondering if the author observed patient-specific signatures (clusters).

Thank you for raising this important point. Of the 7 neutrophil clusters we have identified, none are patient specific i.e. all clusters contain cells from multiple patients. The cluster with the lowest patient number contribution was the exhausted neutrophil population, with representation from 3 patients, and a signal dominated by 2 patients. We have now expanded the results text to clarify this, see page 12 line 301-302.

2. The authors employed riBAQ to perform single-cell data processing (normalization). There is not enough information to explain why such a method is used. Single-cell proteomics is a relatively new field, and this study represents a first-of-its-kind application in a clinical setting. More information will help the field to adopt this method in other disease studies.

We apologise for this oversight. We used riBAQ because it provides a normalised measure of intensity that accounts for the protein size while being more comparable across experiments. Proteins with higher molecular weight produce more theoretical peptides and thus can display higher intensity. The riBAQ methodology addresses this by reducing the intensity of proteins with higher theoretical peptides and increasing it for those with fewer theoretical peptides. This decouples the abundance from the sequence length or molecular weight of the protein. For example, in neutrophils lactoferrin can have the highest total intensity in a proteomics experiment. However, lactoferrin is a large protein of 78,182 Da and many tryptic peptides (~52). S100A9 frequently displays similar intensity to lactoferrin, yet only has 13,342Da and much fewer tryptic peptides (~9). When we estimate protein copy numbers, we see that S100A9 is more abundant than lactoferrin, because a smaller protein showing similar intensity implies higher number of protein copies. When we calculate the riBAQ, instead of just the normal intensity, it shows results that are comparable to the estimated copy numbers, with S100A9 being more abundant. We now provide this additional information in the results section, please see page 12, line 288-295.

3. The data missingness is another well-known challenge in single-cell proteomics. How is the data missingness handled in this study?

We agree with the Reviewer that data missingness is an important challenge. We believe missing values can be biologically relevant, so decided not to impute any values and keep them as 0s. Seurat has the capacity to handle these missing data with sparse matrices.

4. Similarly, most of the volcano plots are plotted with P-value with variable cutoff, likely due to the use of multiple hypothesis testing. To avoid confusion, it is better to plot using the same way (adjusted P value with fixed cutoff).

We agree with the Reviewer that plotting the adjusted p-value would have been a good solution. However, in this case the multiple hypothesis correction used is a Q-value, which represent the FDR measure and not a corrected P-value, hence we considered it was not appropriate to directly plot these Q-values. We had added a red dotted line on all volcano plots which denoted P-values with a corresponding Q-value<0.05, where all points above the red dotted line would be significant. To make this clearer we have now also added a text label to the volcano plots denoting this. See figures 2&3.

5. It is interesting to observe that the lytic NETosis Neutrophils have highly abundant complement and coagulation proteins (blood signatures). Since lytic NETosis Neutrophils lose most of their cytoplasmic and nuclear context in the tumor, it is not clear how these blood proteins get inside of the Neutrophils after NETosis.

This is a very interesting question, one worth pursuing in a future study. Whilst our proteomics workflow does not allow us to distinguish between intracellular proteins and proteins stuck on the neutrophil surface, we propose two ways through which this could occur. Firstly, previous published studies have described a crosstalk between the complement cascade, coagulation factors and NETosis. Neutrophils are known to activate the complement cascade during stimulation, and this has in turn been shown

to induce NETosis (PMID: 25900429). Work from our group has reported changes in surface protein binding resulting in neutrophil platelet aggregate formation (PMID: 33997298). Furthermore, complement proteins have been shown to be found on the surface of activated neutrophils and NETs themselves can act as a scaffold for coagulation factors (PMID: 29572545). A very recent paper by Adrover et al provides direct evidence for NET formation and aggregation with platelets at sites of fibrin deposition resulting in vascular occlusion (PMID: 40670787). Secondly, neutrophils are phagocytes that are also known to engage in macropinocytosis, with previous work from our group showing the uptake of extracellular protein for reuse in protein synthesis (PMID: 33822765). By being embedded in a microenvironment rich in complement and coagulation, these proteins can be internalised. We note that not all proteins are expelled in neutrophil extracellular traps, for example the mitochondrial membrane proteins remain unchanged. Hence it is possible these proteins are internalised pre-NET formation and are not released in NETs. To address this very interesting question on the main manuscript, we have expanded the results and discussion and added additional details about this, please see page 14 lines 339-343; page 20, lines 480-482, 488-490; & page 21 lines 491-492.

6. Check Figure 6G. The labels are quite distinct from those of other figures.

We thank the Reviewer for pointing this out and apologise for this oversight, we have now corrected the label.

7. Ref. 28 is missing

We apologise for this and have now added in the reference and re-numbered accordingly.

Reviewer #2 (Remarks to the Author):

Summary statement

Here, using a small cohort of glioblastoma patients and healthy controls, the authors perform an in-depth characterization of circulating and tumor associated neutrophils (TANs) via bulk, mini-bulk, and single cell proteomic analyses. They clearly outline the sensitivity of each approach and demonstrate quantification of key functional neutrophil proteins down to the single cell level. Flow cytometry sorting based on cell surface marker expression reveals an expanded population of circulating immature, CD10⁻ low density neutrophils (LDNs) in glioblastoma patients. These immature CD10⁻ neutrophils are molecularly distinct from mature CD10⁺ neutrophil subsets and TANs, which molecularly resemble each other, supporting that TANs are likely mature neutrophils. Integration of the mini-bulk data with 277 single TAN proteomes identified seven neutrophil subtypes, functionally classified by differential protein expression. Single cell analysis confirms that TANs are predominantly mature, with a small proportion of immature TANs suggested to be vascular associated. The six mature TAN subtypes exhibit somewhat distinct proteomic signatures linked to cell states, defined by specific protein markers, enabling observable functional trajectories for TAN subtypes. These analyses highlight TAN turnover and posit an ideal molecular signature that could be utilized for therapeutic intervention (engaged). The authors also identify an interesting TAN subtype suggested to be immunosuppressive, warranting further investigation in the context of tumors.

Overall, this manuscript presents a compelling rationale for proteome level characterization of neutrophils in glioblastoma patients. It is a valuable reference and offers a framework to investigate single cell proteomes, even in small cells with low protein input. The integration of single-cell and bulk data is particularly innovative, highlighting the complexity of bulk datasets, incorporating single cell sensitivity measurements, and enabling functional characterization of cell populations. The authors additionally perform an in-depth analysis that allows the proposal of a clinically relevant cell state mechanism that could serve as a model for future single cell studies. Minor comments and clarifications are listed below to enhance the manuscript. One point of curiosity concerns the classification of resected tumors, given changes in WHO glioblastoma criteria during the patient recruitment period. However, these comments do not diminish the overall enthusiasm of the manuscript.

We would like to thank the reviewer for their interest in our work and for recognising the significance of our innovative data integration and generation of a framework in which to study single cell proteomes.

Remarks

1. Additional detail or a diagram related to CD markers and their functional meanings would be useful to provide in the main text or to extended data Figure 2 for those outside of the field.

We thank the Reviewer for their suggestion and agree this would be very useful to make the manuscript more accessible to a wider audience. Hence have included the following text to clarify how CD markers relate to neutrophil function and maturity:

'The evaluated markers were CD10, CD66b and CD11b. CD10 is also known as common acute lymphoblastic leukemia antigen and has previously been identified as a marker of neutrophil maturity²⁸. CD66b, also known as carcinoembryonic antigen-related cell adhesion molecule 8, and CD11b, known as integrin alpha M, are proteins vital for cell adhesion and migration widely used as neutrophil activation markers'.

The text is included within the main manuscript in page 7 lines 169-174. We have also included the following extra references to substantiate the descriptions we included.

PMID: 28053192

PMID: 26970376

PMID: 18056392

PMID: 7773799

PMID: 6480827

2. Why is median and not total summed molecular weight of all detected proteins used for normalization?

This is an interesting question; one we have considered closely. By dividing proteins by their molecular weight and multiplying by the median (47,485 Da) the intensities were kept in virtually the same order of magnitude. Multiplying by the sum of the molecular weights (217,199,587 Da) would have shifted the order of magnitude of

intensity scale considerably. Though both solutions are equally valid, we opted for the option which maintained the intensity scale virtually intact. We have also included additional details regarding this selection within the methods section, page 28, lines 700-701.

3. Regarding extended figure 3, is it interesting that the 25 protein differentiators of all neutrophil subtypes are not differentially expressed between LDN (CD10+) and NDN (CD10+)? Any other thoughts on difference between mature neutrophils with differing densities?

The Reviewer poses a very interesting question. We had expected to discover more differences between LDN (CD10+) and NDN (CD10+), hence the results were surprising to us too. We do not currently have an answer as to why mature neutrophils can differ in density and posit that this could be related to changes in lipid composition. We have added a sentence to this effect in the manuscript, see page 8, line 201-202.

4. Sample stratification and any additional relevant covariates across individual human subjects or sex should be made clear within the figures and/or methods, especially as it relates to the mini-bulk and single cell datasets.

We agree with the Reviewer and have now provided a table detailing patient demographics within the methods section, including subjects used to generate the mini-bulk and single cell datasets, see methods section page 22, line 531.

5. In 2021, during the time period of cohort recruitment, the definition of glioblastoma changed to a very specific classification based on genetic testing and histological classification (PMID: 34185076). Is there any additional depth to tumor classification that can be achieved in these samples? Do the authors know if a patient's tumor misclassification would impact the results? In Figure 2D it looks like there might be a single patient that is an outlier.

We thank the Reviewer for highlighting this important point. All the samples for the mini-bulk and single cell proteomics data were acquired in 2023-2024, hence are based on the new classification and are all IDH1 wildtype. We have also looked back through our more historical data – which only pertains to the flow cytometry data shown on extended figure 2. To prevent any confusion with respect to classification, we have removed the first 4 panels (A-D in the old figure) which featured older samples. All our patients had IDH mutation testing and a histological classification. Beyond this, with our relatively small sample size it is not possible determine whether tumour misclassification would impact the proteomics results or to gain any further depth to tumour classification.

6. Given the frequency of males to females diagnosed with glioblastomas are there any sex differences observed in the neutrophil populations?

The Reviewer poses a very interesting question, unfortunately given our small sample size it is not possible to perform meaningful sex specific analysis of the data.

7. Where is the comparison for CD10+ LDN to CD10- LDN and why is that not the comparison for mature (CD10+ NDN) vs immature (CD10- LDN) neutrophils that only differ by their CD10 expression and not additionally density?

Though the Reviewer is correct that CD10+ LDN would only differ from the CD10- LDN on the density axis, this population is not the default mature (CD10+) neutrophil population that is present in humans. The majority of mature human neutrophils exist in the normal density state (NDN), hence we used this population to compare against immature (CD10- LDN) neutrophils. We did however, in our initial analysis compare CD10+ NDN and CD10+ LDN. As they were so similar we had originally decided not to include the plot comparing CD10+ LDN to the CD10- LDN population within the manuscript. On reflection, this is an important omission and we have now extended the figures to include this comparison. The new plot is included in the new extended figure 3B. Furthermore, to highlight the similarities, we also provide within this document a side-by-side comparison of the volcano plots comparing CD10+ LDN vs CD10- LDN AND CD10+ NDN vs CD10- LDN.

8. Why are LDN CD10+ neutrophils excluded from extended data figure 3 and 4. Is this much variation in detected CD10 expected for CD10+ neutrophils?

As mentioned in the response to point 7, the LDN CD10+ neutrophils were very similar to the NDN CD10+ neutrophils. For simplicity and to avoid redundancy, we had not included them in the figures previously. However, in light of the Reviewer comments, we now include the LDN CD10+ population in the new extended data figure 3B and extended data figure 4.

9. Appreciate the Spectronaut parameters table in the methods.

We thank the Reviewer for their comment.

10. It would be helpful to have specific references that add validity to maturation markers identified from the proteomic mini-bulk comparison of LDN CD10- and NDN CD10+ referred to in lines 213-215. These markers are used frequently to classify

single cell TAN populations in this manuscript, therefore it would be helpful to know exactly how they were selected. The authors should be more upfront if they are referring to maturity and immaturity markers as basically differentially expressed proteins in CD10- comparisons. The markers in Figure 2K, 2L and 4C do not match.

We agree with the Reviewer and apologise for this omission. The majority of the markers we used are known to be associated with neutrophil maturity and have thus added references (PMID: 26970376, PMID: 10947066, PMID: 10614782, PMID: 14730556) substantiating this point (see page 8 line 220). In short, the S100 family of proteins are secretory vesicle proteins, known to be increased in abundance in mature neutrophils (PMID: 10614782). NCF proteins are subunits of the NADPH oxidase, also displaying highest abundance in mature neutrophils (PMID: 10947066). Furthermore, the marker with no published association to maturity, SLC2A3, has been specified as such in the main text (see page 8 lines 220-222).

On page 14, lines 326-330 we have expanded the text to include information and references on PCNA and CEBPE. PCNA is linked to cell cycle and immaturity and CEBPE is a transcription factor also linked to immaturity and production of secondary granules (PMID: 17512402, PMID: 11138622, PMID: 11239167). In addition to this, we have updated Figures 2K-L (now Fig. 2L-M) to match with Figure 4C. We have added NAMPT, PCNA and CEBPE to figure 2L and have labelled Figure 2M and 4C consistently.

11. The utilization of proteins to classify functional neutrophil subtypes is clever. However, a substantial lack of references for the purported functional relationship of each marker protein throughout the results are missing in the manuscript.

We apologise for this oversight and have updated the main text to include additional references which relate to the functional clusters. For immature vascular neutrophil population see page 14 lines 326-330, for lytic NETosis population see page 14 lines 339 & 343, 349, for vital NETs page 15 line 358, 361, for exhausted neutrophils page 15 line 368. We list the added citations below:

PMID: 17512402
PMID: 11138622
PMID: 40356902
PMID: 29572545
PMID: 40670787
PMID: 24009232
PMID: 30898863
PMID: 39300084
PMID: 32663035
PMID: 39697327
PMID: 34870329
PMID: 33306983

Minor comments

12. There is a space needed in line 203

We have corrected this on the main manuscript.

13. For GO enrichment analysis of Figure 2, what is the threshold for significance.

We apologise for not making this clearer. For all GO enrichment analysis all terms required a $FDR < 0.05$ to be considered significant. This has now been specified on the figure legend.